# Air pollution and individuals' mental well-being in the adult population in United Kingdom: A spatial-temporal longitudinal study and the moderating effect of ethnicity

**Mary Abed Al Ahad**[1]*, **Urška Demšar**[1], **Frank Sullivan**[2], **Hill Kulu**[1]

1 School of Geography and Sustainable Development, University of St. Andrews, Scotland, United Kingdom, 2 School of Medicine, University of St. Andrews, Scotland, United Kingdom

* maaa1@st-andrews.ac.uk

**Data Availability Statement:** We cannot make the data underlying our analysis publicly available due

## Abstract

### Background

Recent studies suggest an association between ambient air pollution and mental well-being, though evidence is mostly fragmented and inconclusive. Research also suffers from methodological limitations related to study design and moderating effect of key demographics (e.g., ethnicity). This study examines the effect of air pollution on reported mental well-being in United Kingdom (UK) using spatial-temporal (*between-within*) longitudinal design and assesses the moderating effect of ethnicity.

### Methods

Data for 60,146 adult individuals (age:16+) with 349,748 repeated responses across 10-data collection waves (2009–2019) from "*Understanding-Society: The-UK-Household-Longitudinal-Study*" were linked to annual concentrations of $NO_2$, $SO_2$, PM10, and PM2.5 pollutants using the individuals' place of residence, given at the local-authority and at the finer Lower-Super-Output-Areas (LSOAs) levels; allowing for analysis at two geographical scales across time. The association between air pollution and mental well-being (assessed through general-health-questionnaire-GHQ12) and its modification by ethnicity and being non-UK born was assessed using multilevel mixed-effect logit models.

### Results

Higher odds of poor mental well-being was observed with every 10μg/m³ increase in $NO_2$, $SO_2$, PM10 and PM2.5 pollutants at both LSOAs and local-authority levels. Decomposing air pollution into spatial-temporal (*between-within*) effects showed significant *between*, but not *within* effects; thus, residing in more polluted local-authorities/LSOAs have higher impact on poor mental well-being than the air pollution variation across time within each geographical area. Analysis by ethnicity revealed higher odds of poor mental well-being with increasing concentrations of $SO_2$, PM10, and PM2.5 only for Pakistani/Bangladeshi, other-

to ethical and legal restrictions. We are using the "Understanding Society: The UK Household Longitudinal Study (UKHLS)" dataset which is an initiative funded by the Economic and Social Research Council and various Government Departments, with scientific leadership by the Institute for Social and Economic Research, University of Essex, and survey delivery by NatCen Social Research and Kantar Public. These data are protected by a copyright license and strictly distributed by the UK Data Service which is the largest digital repository for quantitative and qualitative social science and humanities research data in the UK. Therefore, data underlying our analysis can only be accessed through the UK Data Service for authorized researchers from the following URL: https://beta.ukdataservice.ac.uk/datacatalogue/series/series?id=2000053. We confirm that authors did not have any special access privileges that others would not have. The data can be accessed following registration on the UK Data Archive. An application to access geography data such as the local authority and the Lower Super Output Areas is required using the UK Data Archive platform.

**Funding:** This paper is part of a PhD project that is funded by the St Leonard's PhD scholarship, University of St Andrews, Scotland, United Kingdom.

**Competing interests:** The authors declare that they have no conflict of interest

ethnicities and non-UK born individuals compared to British-white and natives, but not for other ethnic groups.

## Conclusion

Using longitudinal individual-level and contextual-linked data, this study highlights the negative effect of air pollution on individuals' mental well-being. Environmental policies to reduce air pollution emissions can eventually improve the mental well-being of people in UK. However, there is inconclusive evidence on the moderating effect of ethnicity.

## 1. Introduction

Mental health problems are rising noticeably world-wide causing serious socio-economic losses to the societies manifested in diminished work productivity and contributing to higher rates of criminal activity and lack of trust in governments [1, 2]. The global burden of mental diseases is estimated at 32% of years lived with disability and 13% of disability-adjusted life-years [3]. Mental disorders are mainly triggered by genetics and/or by psycho-social risk factors [1]. However, recent literature has shown a relationship between environmental factors including exposure to ambient air pollution and mental well-being that ranges from subjective stress and anxiety to more severe depression and suicidal ideation; though most of the evidence is fragmented and inconclusive [1, 4–7].

Ambient air pollution is a mixture of particles (e.g., black carbon and particulate matter with diameters less than or equal to 10 μm: PM10 and to 2.5 μm: PM2.5) and gaseous chemicals (e.g., sulphur dioxide: $SO_2$, nitrogen dioxide: $NO_2$, carbon monoxide: CO, and ozone) that are released to the atmosphere from natural processes (e.g., windblown soil, volcano ashes, pollen, dust) or from man-made activities including energy production, livestock farming, traffic exhaust, and industrial and mining processes [8, 9]. Inhalation of air pollutants can have major consequences on the human central nervous system and neuro-behavioural mechanisms [1, 5, 9, 10]. For example, particulate matter of small diameter such as PM1 or PM2.5 might initiate oxidative stress and lead to the formation of inflammatory cytokines that infiltrate the blood-brain barrier causing neurodegeneration and neuroinflammation [11]. Specifically, exposure to ambient PM2.5 results in depressive responses and increased hippocampal pro-inflammatory cytokines [11], while exposure to PM1 leads to increased inflammation and reactive oxygen species (ROS) generation and impacts learning and memory [12]. In this context, traffic related air pollution (e.g., particulate matter and $NO_2$) have been linked by observational research to increased rates of mental health problems including: autism spectrum disorders [13], schizophrenia [14], dementia [15], psychotic experiences [16, 17], cognitive disabilities [18], anxiety and major depressive disorders [19]. Moreover, higher rates of emergency hospital admissions for depressive disorders have been found on more polluted days in Canada [20–22]; and self-reported mental well-being was associated with long-term exposure to $NO_2$, PM10, and carbon monoxide pollution in Korea [6].

Air pollution can also affect mental well-being indirectly through nuisance and individuals' coping behaviours. Air pollution can result in cognitive anxiety, stress, and loneliness leading to general fatigue and perceived symptoms of poor mental well-being due to aesthetic/odorous nuisance and inhibition of psychological-supporting outdoor activities [23–25]. For example, people may prefer to stay indoors rather than enjoying outdoor activities during periods of high air pollution, especially when air pollution is characterised by visible signs (e.g., colour)

and/or bad odours [26]. In a recent systematic literature review of 178 published articles, air pollution was shown to decrease happiness and life satisfaction substantially, and to increase anxiety, annoyance, mental problems, suicidal ideation, and coping approaches such as avoidance behaviour and migration [2].

Air pollution has been also linked to stress and experiential anxiety emerging from worrying feelings about one's physical health and future [27]. Due to the more conclusive research about the effect of air pollution on physical health including cardiopulmonary, immune system and cancer diseases [8, 9, 28], people living in highly polluted areas might experience stress and worrisome feelings of getting physical illness, which impairs their mental well-being.

Despite the establishment of linkages between air pollution and mental well-being in the literature, results are generally inconclusive and suffer from methodological drawbacks related to the chosen study design and methods of estimating air pollution [16]. Most of the studies are either cross-sectional studies or longitudinal studies that lack spatial-temporal specificity and a lengthy follow-up time. To date, no study has tried to address the association between long-term (11 years) air pollution exposure and mental well-being using a spatial-temporal *(between-within)* longitudinal design. A *between-within* analysis can determine the *spatial time-constant* cross-sectional (average 11 years air pollution) effect of air pollution on mental well-being *between* different geographical areas (e.g., local authorities or census output areas) as well as the *temporal time-varying* longitudinal (yearly air pollution deviation from the 11 years average) effect of air pollution on mental well-being *within* each geographical area. In other words, this analysis reveals whether living in more polluted local authorities or census output area is the driving cause for poor mental well-being (*between*) or whether it is the fluctuation in air pollution across time within each local authority or census output area (*within*) that is causing poor mental well-being. This type of analysis combines both cross-sectional and longitudinal designs; thus establishing a more robust measurement of the effect of air pollution [29].

Furthermore, published research has not yet covered population subgroups and the potential moderating effect that key demographic groups might have on the association between air pollution and mental well-being. To date, only age (young vs elderly population) and gender have been reported as effect modifiers for the association between air pollution and mental well-being. In Korea, a more pronounced risk of stress and depression from air pollution exposure was observed among men than among women and among people aged less than 65 [6]. In China, exposure to increased concentrations of ambient air pollution showed a greater risk for mental health problems and general well-being among female college students [30]. Therefore, examining how the effect of air pollution on mental well-being varies by other key demographic characteristics such as ethnicity can provide a more conclusive explanation for the association between air pollution and mental well-being. Earlier literature has shown that ethnic minorities suffer from relatively higher levels of stress, depression, and self-harm [31, 32]. This could be attributed to their lower socio-economic status and to living in more deprived ethnic communities with poor housing and neighbourhood conditions [33–35]. Ethnicity was also examined in the literature from the lens of migration and being a non-native resident. A systematic literature review in Sweden showed increased risks of depression and psychotic problems among immigrants compared to the native population [36]. Given the higher observed risk of mental problems among ethnic minorities and immigrants, the effect of air pollution on mental well-being among ethnic groups should be investigated.

Accordingly, this study investigates longitudinally the overall and the spatial-temporal *(between-within)* effects of long-term (11 years) exposure to $NO_2$, $SO_2$, PM10, and PM2.5 air pollution in the UK on individuals' reported mental well-being measured using the 12 items General Health Questionnaire (GHQ12) scale. Unlike other studies that assess the effect of air

pollution on well-being using one geographical scale, our study aims to assess the effect of air pollution exposure on mental well-being at two geographical scales, coarse local authorities (council areas in Scotland) and detailed Lower Super Output Areas (LSOAs; data zones in Scotland). This will allow us to compare the results between the two geographical scales and explore in more detail the local-contextual patterns of the effect of air pollution on mental well-being. Additionally, our study aims to consider both space and time by determining whether living in more polluted geographical areas (local authorities and LSOAs) is the driving cause for poor mental well-being (*between*) or whether it is the variation in air pollution across time within each geographical area (*within*) that is causing poor mental well-being; thus providing detailed spatial-temporal evidence for policymaking decisions. Finally, we aim to investigate whether ethnic minorities including Pakistani/Bangladeshi, Indians, Black/African/Caribbean, mixed and other ethnicities as well as non-UK born individuals suffer from a more pronounced risk for mental well-being with increasing concentrations of the four air pollutants compared to British-White ethnicity and UK-born individuals, respectively.

## 2. Materials and methods

### 2.1. Study design and population

This study utilises individual-level data from the "*Understanding Society*: *The UK Household Longitudinal Study (UKHLS)*" [37]. The UKHLS is a rich longitudinal dataset that is composed of 10 data collection panels/waves over a period of 12 years from (2009–2020) with about 40,000 households enrolled at the first wave from the four UK nations: England, Wales, Scotland, and Northern Ireland.

The dataset contains yearly information on the self-reported general health and mental well-being of individuals and on their socio-demographic characteristics including gender, age, educational qualification, marital status, occupation, perceived financial situation, ethnicity, and country of birth. In addition, the dataset collects yearly information on individuals' lifestyle factors such as cigarette smoking and on contextual factors such as the local authority/council area and the Lower Super Output Areas (LSOAs)/data zones where households are located [37].

The UKHLS main survey sample is composed of four data collection sub-samples as summarised in Fig 1 and described in detail in other publications [37–39].

The present study utilises individual-level data from the UKHLS survey on 60,146 adult (age: 16+) individuals with 349,748 repeated responses (at least 2 repeated responses per individual) collected within 10 waves over a period of 11 years (2009–2019). It should be noted, however, that the initial UKHLS adult survey includes a total of 87,045 individuals with

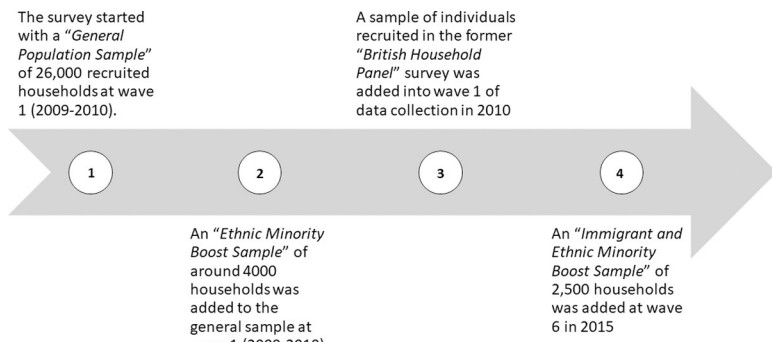

**Fig 1. A diagram showing the four data collection sub-samples of the UKHLS main survey.**

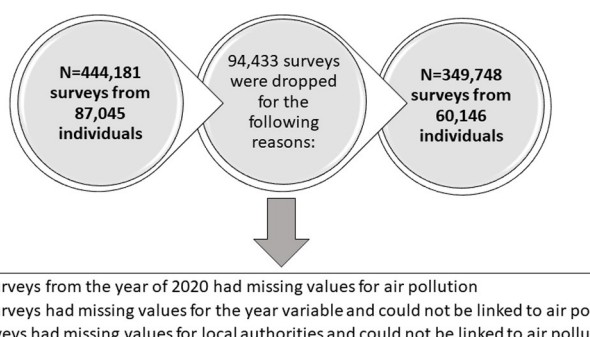

**Fig 2. A diagram showing the reasons for dropping survey responses from the UKHLS longitudinal panel data.**

444,181 repeated responses. We dropped 94,433 observations due to the reasons described in Fig 2.

## 2.2. Variables and measurements

**2.2.1. Individuals' reported mental well-being.** Individuals' reported mental well-being was measured using the 12-items "General Health Questionnaire (GHQ12)" scale [40] which is widely used in the population health research to capture non-psychotic psychiatric illness. The GHQ12 scale is composed of 12 questions about the individuals' experience of 12 symptoms related to mental well-being in the past few weeks preceding data collection date. The 12 questions are: 1) Ability to concentrate; 2) Losing much sleep; 3) Playing a useful part; 4) Capability of taking decisions; 5) Being under stress; 6) Inability to overcome difficulties; 7) Enjoying normal activities; 8) Being able to face up problems; 9) Feeling unhappy and depressed; 10) Losing confidence; 11) Thinking of self as worthless; and 12) Feeling reasonably happy [41]. Individuals are then asked to rate the negative questions as 0 = 'not at all', 1 = 'no more than usual', 2 = 'rather more than usual', 3 = 'much more than usual' and the positive questions as 0 = 'better than usual', 1 = 'same as usual', 2 = 'less than usual', and 3 = 'much less than usual' [42]. Two methods are used by relevant literature to construct the overall GHQ12 well-being score. The first and most used method is the (0-0-1-1) method whereby responses for each of the 12 questions of the GHQ12 scale are dichotomised (0 and 1 into 0; 2 and 3 into 1) and then the 12 items are summed up resulting in a general mental well-being score ranging from 0 to 12 with higher scores indicating poorer mental well-being [42–46]. The second method aims to construct a Likert scale score (0-1-2-3) by adding up all the 12 items of the GHQ12 scale resulting in a total mental well-being score ranging from 0 to 36 with higher scores indicating poorer mental well-being [47, 48].

For the present study, we used both methods for the GHQ12 scale, the (0-0-1-1) and the (0-1-2-3) method. Given that the scores of the GHQ12 scale are right-skewed and based on relevant literature, we dichotomised the overall GHQ12 scale using two cut off points for the GHQ12 (0–12): our sample mean GHQ12 (0–12) score = 1.8 ∼ 2 (GHQ0-12 ≥ 2) and the GHQ12 (0–12) score of greater than or equal to 4 (GHQ0-12 ≥ 4) as an indication of poor mental well-being [43–46, 49, 50]. The GHQ12 (0–36) score was dichotomised based on one cut off point of greater than or equal to 12 (GHQ0-36 ≥ 12) as an indication of poor mental well-being [47, 48]. Therefore, GHQ0-12 and GHQ0-36 total scores were treated as binary

variables: *GHQ0-12 ≥ 2* (0 = scores <2, good mental well-being; 1 = scores ≥2, poor mental well-being); *GHQ0-12 ≥ 4* (0 = scores <4, good mental well-being; 1 = scores ≥4, poor mental well-being); and *GHQ0-36 ≥ 12* (0 = scores <12, good mental well-being; 1 = scores ≥12, poor mental well-being).

**2.2.2. Air pollution data.** Raster data of annual mean concentrations of $NO_2$, $SO_2$, PM10, and PM2.5 air pollutants available up to the year of 2019, measured in μg/m³, and estimated using air dispersion models at a 1x1 km spatial resolution were downloaded from the "Department for Environment Food and Rural Affairs (DEFRA)" online data repository [51]. These air pollution maps at 1x1 km resolution are modelled each year by DEFRA under the "Defra's Modelling of Ambient Air Quality (MAAQ) contract" and are used to provide policy support in the UK and to fulfil the UK's reporting obligations to Europe [51]. The 1x1 km air pollution raster data are the finest spatial resolution data that can be downloaded from DEFRA and are sufficient to obtain good modelling estimates [52, 53].

These data were first projected using the UK National Grid projection system in ArcGIS Pro software. Then for each of the 391 local authorities/council areas in the UK, we computed the average concentrations of $NO_2$, $SO_2$, PM10, and PM2.5 pollutants from all the 1x1 km raster cells that fell within the boundaries of the respective local authorities/council areas for each year from 2009 up to 2019. Next, we linked the average concentrations of air pollution at the local authority level to the UKHLS dataset for each individual and each year of follow up (2009–2019).

To minimise exposure bias and establish more robust results from a spatial perspective, we also linked the 1x1 km raster air pollution data to the UKHLS data at the level of Lower Super Output Areas (LSOAs; data zones for Scotland and Super Output Areas for Northern Ireland), a finer geographical scale, for each individual and each year of follow up (2009–2019). LSOAs are used to decompose England and Wales based on the population size into areas with a minimum population size of 1000 people and are the lowest level of geography offered by the UKHLS dataset. The LSOAs in England and Wales are equivalent to data zones in Scotland and to Super Output Areas in Northern Ireland. For simplicity we refer to the joint LSOAs, data zones, and Super Output Areas as LSOAs. Using these smaller spatial units, we ran our analysis at a smaller geographic scale than local authorities, which allowed us to explore local patterns of the effect of air pollution.

A map showing the local authorities in the UK (council areas in Scotland) and an enlarged subset of 20 local authorities in the south-east of UK with an example of PM10 concentrations at 1x1 km resolution for the year of 2017 for Tower Hamlets local authority and its corresponding LSOAs is used to clarify the process of linking air pollution to the UKHLS dataset at the two geographical scales, the coarse local authorities and the detailed LSOAs (Fig 3).

**2.2.3. Socio-demographic and lifestyle covariates.** A list of individual-level socio-demographic and lifestyle covariates summarised in Table 1 was selected a priori for this study based on what is available in the UKHLS dataset and based on the potential confounders considered by relevant literature. Most of the researchers in the field of air pollution and mental well-being have considered age [4, 6, 16], gender [2, 4, 6, 16], education [4, 6], marital status [6], socio-economic deprivation and occupation [16, 56], ethnicity [1, 7, 16], cigarette smoking [16, 56], alcohol drinking [4, 16], physical exercise [4, 16], and body mass index [4] as potential confounders. Specifically, poor mental well-being, stress, and depression have been associated with younger or older ages, women, cigarette smoking, alcohol drinking, physical inactivity, lower education, divorced/widowed individuals, lower household income, and belonging to an ethnic minority group [6, 31–33], which in turn confounds the association between air pollution and mental well-being outcomes. In addition, environmental-contextual factors such as noise pollution [1, 16], neighbourhood disorder [16], green spaces availability [25, 57], air

**Fig 3. A map showing the local authorities in the UK and an enlarged subset of 20 local authorities in the south-east of UK with an example of PM10 concentrations at 1x1 km grid for the year of 2017 for Tower Hamlets local authority and its corresponding LSOAs.** The map was constructed by the authors in ArcGIS Pro software using PM10 air pollution shapefile for the year of 2017 downloaded from the DEFRA online data repository [51], local authorities UK boundaries shapefile downloaded from the Office for National Statistics [54], and LSOAs and data zones UK boundaries also downloaded from the Office for National Statistics, National Records of Scotland, and Northern Ireland Statistics [55]. Both DEFRA and Office for National Statistics shapefiles are governed under the Open Government Licence v.3.0.

temperature [57], and seasonality [16, 58] have been also considered as potential confounders in the association between air pollution and mental well-being.

## 2.3. Data analysis

Individuals' socio-demographic and lifestyle factors were described using percentages for each wave (waves 1 to 10) of the UKHLS sample.

The mean of $NO_2$, $SO_2$, PM10, and PM2.5 concentrations and the Pearson's correlation between the four pollutants was computed at both geographical scales, the coarse local authorities and the detailed LSOAs. Given the high observed correlations between $NO_2$, PM10, and PM2.5 pollutants (Pearson's coefficient $\geq$ 0.7 [59]; Tables 3 and 4), the association of $NO_2$, $SO_2$, PM10, and PM2.5 with individuals' reported mental well-being was analysed in separate regression models. Nevertheless, low to moderate correlation was observed between $SO_2$ and each of the other three pollutants which enabled the construction of bi-pollutant models adjusting the $NO_2$, PM10, and PM2.5 models for the $SO_2$ pollutant.

Intraclass correlation coefficients (ICCs) were also calculated to assess the homogeneity in the mental well-being scores within individuals and household clusters; with an ICC of greater than 0.3 indicating moderate to fair homogeneity [60]. Given the presence of 42% and 49% homogeneity (ICC = 0.42 and ICC = 0.49; Table 5) for the mental well-being scores of GHQ0-12 and GHQ0-36 within individuals' clusters across time, respectively, the mean score of mental well-being was computed from predictions of mixed effects linear models, adjusting for age in fixed effects and for the individual ID in random intercept.

The association between the three binary measures (GHQ0-36 $\geq$ 12; GHQ0-12 $\geq$ 2; GHQ0-12 $\geq$ 4) of individuals' reported mental well-being and each of $NO_2$, $SO_2$, PM10, and PM2.5 pollutants (linked at two geographical scales: once at the LSOAs and once at the local authorities level) was examined separately and in bi-pollutant models adjusted for $SO_2$ using three-levels (repeated individual observations across time nested within LSOAs or local authorities) mixed-effect logit models, adjusting for the socio-demographic and lifestyle covariates and for the year (2009–2019) dummies. This type of analysis was chosen as it fits the longitudinal panel design of the study which involves repeated individual responses across time linked to air pollution data at the LSOA or local authority level whereby repeated individual responses are nested within LSOAs or local authorities. The individual-level random intercept

is necessary in the multilevel models given the high homogeneity in the individual's responses across time (ICC = 0.42 and ICC = 0.49; Table 5), while the local authority or LSOAs random intercept is needed to allow for less biased assessments of the contextual-linked air pollution effect on mental well-being [61]; resulting in three-levels mixed-effect logit models. In a supplementary analysis, we also show the association of individuals' reported mental well-being with each of the socio-demographic and lifestyle covariates (Table 1 in S1 File). It should be noted, however, that our models did not account for the household clustering in random intercepts due to the low observed homogeneity in the well-being responses within each household cluster (ICC = 0.16 and ICC = 0.18; Table 5).

In additional analysis, we decomposed the overall effect of air pollution (linked at two geographical scales: LSOAs and local authorities) on mental well-being into *between* (*spatial*) and *within* (*temporal*) effects. *Between* effects were used to examine the *spatial time-constant* effect

**Table 1. The socio-demographics and lifestyle covariates selected for this study.**

| Covariates | Coding |
|---|---|
| Gender | 1 = Male<br>2 = Female |
| Age | Coded as 16–18 and then in 5 years increments (19–23; 24–28; 29–33; 34–38; 39–43; 44–48; 49–53; 54–58; 59–63; 59–63; 64–68; 69–73; 74–78; >78) |
| Ethnicity | 1 = British white<br>2 = Other white<br>3 = Indian<br>4 = Pakistani/Bangladeshi<br>5 = Black/African/Caribbean<br>6 = Mixed ethnicities<br>7 = Other ethnicities |
| Country of birth | 1 = Born in UK<br>2 = Not born in UK<br>3 = No answer |
| Marital status | 1 = Married<br>2 = Living as a couple<br>3 = Widowed<br>4 = Divorced/separated<br>5 = Single never married<br>6 = No answer |
| Educational qualification | 1 = University degree<br>2 = High school degree<br>3 = Lower educational qualifications<br>4 = Other qualifications<br>5 = Still a student |
| Occupation | 1 = Managers/Professionals/employers<br>2 = Non manual workers<br>3 = Manual workers<br>4 = Not applicable: Student/retired/Not working<br>5 = No answer |
| Perceived financial situation | 1 = Living comfortably/doing alright<br>2 = Living difficultly<br>3 = No answer |
| Cigarette smoking[a] | 0 = Non-smoker<br>1 = Smoker<br>2 = No answer |

Cigarette smoking[a]: data for cigarette smoking were missing for wave 1 responses and for waves 3 and 4 among individuals aged more than 21 years. Therefore, we replicated the smoking status responses of wave 2 in wave 1 for each individual and in waves 3 and 4 for each individual aged more than 21 years. The smoking status in waves 1, 3, and 4 was coded into "no answer" for individuals who were not present in wave 2.

of air pollution *between* different geographical areas (local authorities and LSOAs) by calculating the average concentrations of $NO_2$, $SO_2$, PM10 and PM2.5 pollutants across the 11 years of follow up (2009–2019) for each local authority and each LSOA. On the other hand, *within* effects were used to assess the *temporal time-varying* effect of air pollution *within* each geographical area by calculating the annual air pollution deviation from the 11 years average air pollution for each local authority and LSOA. Therefore, two sets of four multilevel mixed effect logit models (one for each pollutant) were used to examine the overall (Eq 1) and the *between-within* (Eq 2) effects of air pollution on individuals' mental well-being, respectively, at the two geographical scales (coarse local authorities and detailed LSOAs).

$$\ln\left(\frac{Y_{tij}}{1 - Y_{tij}}\right)$$
$$= \beta_0 + U_{0ij} + U_{0j} + \beta_1 \text{overall pollutant concentration}_{tij} + \beta_2 \text{Age}_{tij} + \beta_3 \text{Gender}_{tij}$$
$$+ \beta_4 \text{Ethnicity}_{tij} + \beta_5 \text{Country of birth}_{tij} + \beta_6 \text{Marital status}_{tij} + \beta_7 \text{Education}_{tij}$$
$$+ \beta_8 \text{Occupation}_{tij} + \beta_8 \text{Perceived financial situation}_{tij} + \beta_9 \text{Smoking status}_{tij}$$
$$+ \beta_{10} \text{Year dummies}_{ij} \tag{1}$$

$$\ln\left(\frac{Y_{tij}}{1 - Y_{tij}}\right)$$
$$= \beta_0 + U_{0ij} + U_{0j} + \beta_1 \text{Between pollutant concentration}_{tij}$$
$$+ \beta_2 \text{Within pollutant concentration}_{tij} + \beta_3 \text{Age}_{tij} + \beta_4 \text{Gender}_{tij} + \beta_5 \text{Ethnicity}_{tij}$$
$$+ \beta_6 \text{Country of birth}_{tij} + \beta_7 \text{Marital status}_{tij} + \beta_8 \text{Education}_{tij} + \beta_9 \text{Occupation}_{tij}$$
$$+ \beta_{10} \text{Perceived financial situation}_{tij} + \beta_{11} \text{Smoking status}_{tij} + \beta_{12} \text{Year dummies}_{ij} \tag{2}$$

Where $Y_{tij}$ is the mental well-being outcome for individual *i*, in local authority or LSOA *j* at year *t*; $\beta_1, \beta_2 \ldots \beta_{12}$ are the slopes of fixed effects; $\beta_0$ is the fixed intercept; $U_{0ij}$ is level 2 random intercept of individuals nested in local authorities or LSOAs; $U_{0j}$ is level 3 random intercept of local authorities or LSOAs.

To assess the effect modification of ethnicity and country of birth on the association between air pollution and mental well-being, we added interaction terms between ethnicity and each of $NO_2$, $SO_2$, PM10, and PM2.5 pollutants and between country of birth and each of the four pollutants. These interaction terms were added into the models investigating the overall effect of air pollution and into the models examining the *between* and *within* effects of air pollution at the two geographical scales of local authorities and LSOAs, separately (i.e., one interaction term at a time, once with the *between* effect and once with the *within* effect). The interaction results were visualised with coefficient plots.

To estimate the cohort effect and to balance the follow up time, we repeated the same multilevel mixed effect logit modelling in a sensitivity analysis only for individuals recruited during the first wave of the UKHLS.

STATA software (StataCorp. 2015. Stata Statistical Software: Release 14. College Station, TX: StataCorp LP) was used for statistical analysis and ArcGIS Pro software was used for spatial pre-processing of air pollution data. Odd ratios (ORs) and 95% confidence (CIs) for every 10 $\mu g/m^3$ increase in air pollution were used to report the study findings. Statistical significance was considered at a P-value < 0.05.

## 2.4. Ethical considerations

This paper was granted ethical approval on the 14th of May 2020 by the authors' affiliated institution (School of Geography and Sustainable Development Ethics Committee, acting on behalf of the University Teaching and Research Ethics Committee (UTREC) at the University of St Andrews). The paper uses secondary adult (age 16+) fully anonymised data from the "Understanding Society: The UK Household Longitudinal Study (UKHLS)" and authors did not have access to potentially identifying information; thus, obtaining participants' informed consent is not applicable and was waved by the authors' institution ethics committee. The University of Essex responsible for the UKHLS data collection and management has already obtained written informed consent from all the study participants [37]. Requesting consent for health record linkage was approved at Wave 1 by the National Research Ethics Service (NRES) Oxfordshire REC A (08/H0604/124), and at Wave 4 by NRES Southampton REC A (11/SC/0274). Approval for the collection of biosocial data by trained nurses in Waves 2 and 3 of the main survey was obtained from the National Research Ethics Service (Understanding Society—UK Household Longitudinal Study: A Biosocial Component, Oxfordshire A REC, Reference: 10/H0604/2).

## 3. Results

### 3.1. Individuals' socio-demographic and lifestyle characteristics

A total of 60,146 adult individuals with 349,748 repeated responses over 11 years (2009–2019) of 10 data collection waves were included in this study. The mean of observations per individual was 5.81 (SD = 2.81) with a minimum of 2 observations per individual and the average follow up time was 5.58 (SD = 2.98) years.

Description of the individuals' socio-demographic and lifestyle characteristics for the 10 data collection waves of the UKHLS are summarised in Table 2. For all waves, the majority of individuals were females, belonged to the middle-aged group (34–58 years), were married, had

**Table 2. Description of individual's socio-demographic and lifestyle factors for each wave of the UKHLS data (N = 349,748 surveys from 60,146 individuals).**

| | | Wave1 (2009–2011) | Wave2 (2010–2012) | Wave3 (2011–2013) | Wave4 (2012–2014) | Wave5 (2013–2015) | Wave6 (2014–2016) | Wave7 (2015–2017) | Wave8 (2016–2018) | Wave9 (2017–2019) | Wave10 (2018–2019) |
|---|---|---|---|---|---|---|---|---|---|---|---|
| | | N = 31,258 | N = 39,858 | N = 38,632 | N = 37,315 | N = 35,190 | N = 36,349 | N = 35,572 | N = 34,348 | N = 31,741 | N = 29,485 |
| Gender | Male | 43.2% | 43.6% | 43.8% | 44.0% | 44.3% | 44.3% | 44.4% | 44.4% | 44.3% | 44.0% |
| | Female | 56.8% | 56.4% | 56.2% | 56.0% | 55.7% | 55.7% | 55.6% | 55.6% | 55.7% | 56.0% |
| Age | Young (<34) | 26.2% | 25.4% | 25.8% | 25.2% | 25.1% | 24.6% | 24.0% | 23.3% | 22.5% | 20.8% |
| | Middle age (34–58) | 46.1% | 44.8% | 45.3% | 44.7% | 44.5% | 44.7% | 44.3% | 43.7% | 43.4% | 43.5% |
| | Old (>58) | 27.7% | 29.7% | 28.9% | 30.1% | 30.4% | 30.7% | 31.6% | 33.0% | 34.2% | 35.7% |
| Ethnicity | British white | 81.4% | 82.5% | 81.8% | 81.8% | 82.0% | 76.6% | 76.5% | 76.9% | 78.2% | 79.1% |
| | Other white | 4.1% | 4.8% | 4.7% | 4.6% | 4.4% | 5.6% | 5.5% | 5.4% | 5.3% | 5.1% |
| | Indian | 3.1% | 2.5% | 2.6% | 2.6% | 2.6% | 3.8% | 3.9% | 3.9% | 3.7% | 3.6% |
| | Pakistani/ Bangladeshi | 3.5% | 2.7% | 3.0% | 3.0% | 3.1% | 4.6% | 4.7% | 4.8% | 4.6% | 4.5% |
| | Black/African/ Caribbean | 4.0% | 3.0% | 3.4% | 3.3% | 3.3% | 4.3% | 4.4% | 4.0% | 3.6% | 3.3% |
| | Mixed ethnicities | 1.6% | 1.3% | 1.5% | 1.5% | 1.6% | 1.8% | 1.8% | 1.8% | 1.8% | 1.7% |
| | Other ethnicities | 2.3% | 3.1% | 3.1% | 3.0% | 3.0% | 3.3% | 3.2% | 3.1% | 2.8% | 2.8% |

(*Continued*)

**Table 2.** (Continued)

| | | Wave1 (2009–2011) | Wave2 (2010–2012) | Wave3 (2011–2013) | Wave4 (2012–2014) | Wave5 (2013–2015) | Wave6 (2014–2016) | Wave7 (2015–2017) | Wave8 (2016–2018) | Wave9 (2017–2019) | Wave10 (2018–2019) |
|---|---|---|---|---|---|---|---|---|---|---|---|
| | | N = 31,258 | N = 39,858 | N = 38,632 | N = 37,315 | N = 35,190 | N = 36,349 | N = 35,572 | N = 34,348 | N = 31,741 | N = 29,485 |
| Country of birth | Born in the UK | 86.3% | 67.7% | 67.0% | 68.0% | 68.6% | 66.3% | 66.7% | 67.1% | 68.2% | 68.4% |
| | Not born in the UK | 13.7% | 10.5% | 10.5% | 10.5% | 10.3% | 14.3% | 14.3% | 13.9% | 12.6% | 12.0% |
| | No answer | 0.0% | 21.8% | 22.4% | 21.5% | 21.1% | 19.5% | 18.9% | 19.0% | 19.2% | 19.6% |
| Marital status | Married | 53.2% | 53.6% | 52.7% | 52.1% | 51.7% | 52.9% | 52.8% | 53.2% | 53.8% | 55.2% |
| | Living as a couple | 11.8% | 11.5% | 11.6% | 11.8% | 11.6% | 10.8% | 10.7% | 10.5% | 10.0% | 9.6% |
| | Widowed | 5.5% | 5.9% | 5.7% | 5.8% | 5.8% | 5.7% | 5.7% | 5.8% | 5.9% | 5.9% |
| | Divorced/ separated | 9.1% | 8.4% | 8.5% | 8.6% | 8.5% | 8.1% | 8.0% | 7.9% | 7.9% | 8.1% |
| | Single never married | 20.4% | 20.6% | 21.7% | 21.5% | 22.2% | 22.2% | 22.6% | 22.5% | 22.1% | 20.8% |
| | No answer | 0.1% | 0.0% | 0.0% | 0.1% | 0.2% | 0.4% | 0.2% | 0.2% | 0.3% | 0.4% |
| Educational qualification | University degree | 31.9% | 25.5% | 27.2% | 28.1% | 29.3% | 29.8% | 30.6% | 31.6% | 32.8% | 34.3% |
| | High school degree | 32.9% | 25.7% | 26.2% | 26.5% | 26.7% | 26.2% | 26.4% | 26.7% | 27.0% | 26.9% |
| | Lower educational levels | 1.4% | 1.1% | 1.1% | 1.1% | 1.1% | 1.0% | 1.0% | 1.0% | 1.0% | 1.0% |
| | Other qualifications | 27.4% | 40.8% | 38.7% | 37.6% | 36.3% | 36.5% | 35.5% | 34.7% | 33.5% | 33.3% |
| | Still a student | 6.3% | 6.9% | 6.8% | 6.7% | 6.6% | 6.5% | 6.5% | 5.9% | 5.6% | 4.6% |
| Occupation | Managers/ Professionals/ employers | 12.4% | 12.1% | 12.0% | 12.3% | 12.4% | 12.1% | 12.1% | 12.2% | 12.1% | 11.9% |
| | Non manual workers | 27.5% | 27.1% | 27.6% | 27.3% | 28.0% | 27.5% | 27.7% | 27.1% | 26.8% | 26.5% |
| | Manual workers | 17.9% | 17.9% | 18.2% | 17.9% | 18.3% | 18.2% | 18.3% | 17.8% | 17.2% | 16.3% |
| | Not applicable: Student/ retired/Not working | 42.0% | 42.7% | 41.6% | 42.1% | 40.9% | 41.6% | 41.3% | 41.9% | 41.8% | 42.6% |
| | No answer | 0.2% | 0.2% | 0.7% | 0.4% | 0.3% | 0.6% | 0.6% | 1.0% | 2.2% | 2.7% |
| Perceived financial situation | living comfortably/ doing alright | 59.9% | 62.3% | 62.0% | 64.5% | 66.3% | 70.8% | 72.4% | 73.1% | 71.7% | 71.6% |
| | living difficultly | 40.0% | 37.6% | 37.9% | 35.4% | 33.6% | 29.0% | 27.5% | 26.7% | 28.1% | 28.2% |
| | no answer | 0.1% | 0.1% | 0.1% | 0.1% | 0.1% | 0.3% | 0.2% | 0.2% | 0.2% | 0.2% |
| Cigarette smoking | non-smoker | 73.8% | 79.2% | 70.7% | 69.3% | 82.1% | 77.3% | 84.4% | 85.3% | 86.6% | 87.0% |
| | smoker | 19.6% | 20.8% | 19.0% | 18.6% | 17.9% | 15.6% | 15.5% | 14.6% | 13.3% | 12.8% |
| | no answer | 6.6% | 0.1% | 10.4% | 12.1% | 0.0% | 7.1% | 0.1% | 0.1% | 0.1% | 0.2% |
| Nation | England | 83.1% | 75.6% | 75.8% | 76.1% | 76.9% | 78.7% | 78.7% | 78.7% | 78.2% | 78.3% |
| | Wales | 5.0% | 7.8% | 7.9% | 7.8% | 7.5% | 6.6% | 6.4% | 6.5% | 6.6% | 6.5% |
| | Scotland | 7.4% | 9.5% | 9.5% | 9.3% | 9.5% | 8.5% | 8.6% | 8.5% | 8.7% | 8.8% |
| | Northern Ireland | 4.4% | 7.1% | 6.7% | 6.8% | 6.1% | 6.3% | 6.2% | 6.2% | 6.5% | 6.4% |

**Table 3. Correlation matrix of air pollutants at the LSOAs level (N = 42,619 LSOAs).**

| | NO$_2$ (µg/m$^3$) | SO$_2$ (µg/m$^3$) | PM10 (µg/m$^3$) | PM2.5 (µg/m$^3$) |
|---|---|---|---|---|
| NO$_2$ (µg/m$^3$) | 1.00 | | | |
| SO$_2$ (µg/m$^3$) | 0.37 | 1.00 | | |
| PM10 (µg/m$^3$) | **0.76** | 0.28 | 1.00 | |
| PM2.5 (µg/m$^3$) | **0.79** | 0.32 | **0.97** | 1.00 |

**Table 4. Correlation matrix of air pollutants at the local authority level (N = 391 local authorities).**

| | NO$_2$ (µg/m$^3$) | SO$_2$ (µg/m$^3$) | PM10 (µg/m$^3$) | PM2.5 (µg/m$^3$) |
|---|---|---|---|---|
| NO$_2$ (µg/m$^3$) | 1.00 | | | |
| SO$_2$ (µg/m$^3$) | 0.50 | 1.00 | | |
| PM10 (µg/m$^3$) | **0.77** | 0.38 | 1.00 | |
| PM2.5 (µg/m$^3$) | **0.81** | 0.42 | **0.97** | 1.00 |

**Table 5. Intraclass correlation coefficient for within individual and household clusters.**

| | | Mental well-being GHQ12 scale (0–12) | Mental well-being GHQ12 scale (0–36) |
|---|---|---|---|
| **Individual ID** | ICC [95%CI] | *0.42 [0.41, 0.42]* | *0.49 [0.48, 0.49]* |
| | N of surveys | 349,748 | |
| | N of individuals | 60,146 | |
| | Mean$^a$ (SD) | 1.81 (0.21) | 11.14 (0.05) |
| **Household ID** | ICC [95%CI] | 0.16 [0.16, 0.17] | 0.18 [0.18, 0.19] |
| | N of surveys | 349,748 | |
| | N of households | 217,009 | |

Moderate to fair ICCs>0.3 are highlighted in italic-bold; Mean$^a$ is based on predictions from mixed effects linear models which are adjusted for age in fixed effects and for the individual ID in random intercept.

either a university or high school degree, were non-manual workers (if working), were living comfortably/doing alright financially, and were cigarette non-smokers (Table 2).

For ethnicity, most individuals were UK-born (86% in wave 1) and belonged to the British-white group (81% in wave 1). The description of other ethnic groups in wave 1 is as follows: Other-white (4%), Indians (3%), Pakistani/Bangladeshi (3.5%), Black/African/Caribbean (4%), mixed ethnicities (1.6%), and other ethnicities (2%) (Table 2).

## 3.2. Description of air pollution

**3.2.1. Description of air pollution at the LSOAs level.** Fig 4 shows the average yearly concentrations of NO$_2$, SO$_2$, PM10, and PM2.5 pollutants across the 42,619 LSOAs in the UK from 2009 to 2019. Air pollution showed fluctuations across time with lower concentrations seen in the last 5 years (2015–2019) of observation compared to previous years for all four pollutants (Fig 4).

We also observed high correlations (Pearson's coefficient ≥ 0.7) between NO$_2$, PM10, and PM2.5 pollutants (Table 3), which could be attributed to the source of emission and the atmospheric chemical reactions between these pollutants. For example, the major source of NO$_2$

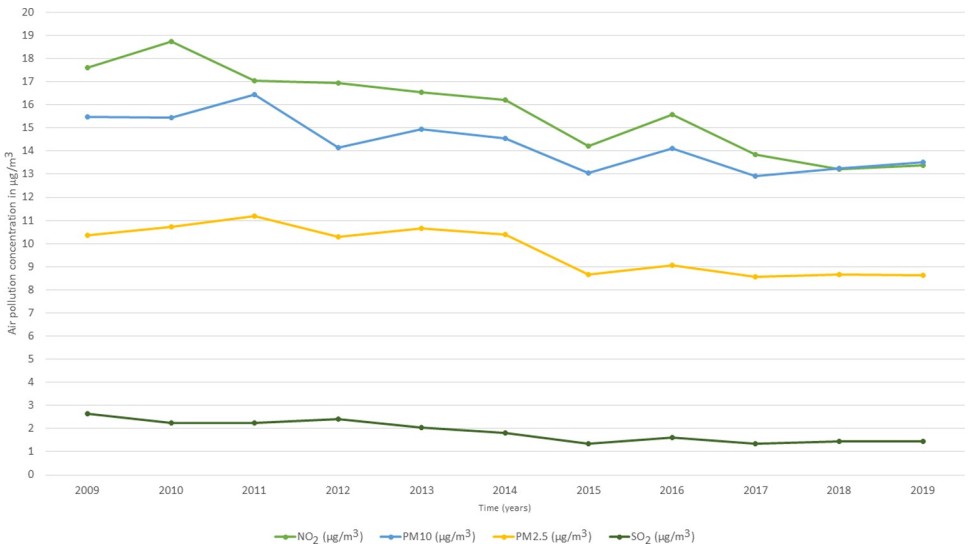

**Fig 4. The annual mean of NO$_2$, SO$_2$, PM10, and PM2.5 air pollutants at the LSOAs level in the UK from the year of 2009 to 2019 (N = 42,619 LSOAs).**

and particulate matter (PM10 and PM2.5) emissions is traffic exhaust [62, 63], while industrial processes and power plants are the major sources of SO$_2$ pollution [64].

**3.2.2. Description of air pollution at the local authority level.** Similar to the air pollution at the LSOAs level, air pollution at the local authority level also showed fluctuations across time with lower concentrations seen in the last 5 years (2015–2019) of observation compared to previous years (2009–2014) for all four pollutants (Fig 5). Likewise, high correlations (Pearson's coefficient ≥ 0.7) between NO$_2$, PM10, and PM2.5 pollutants were noticed (Table 4).

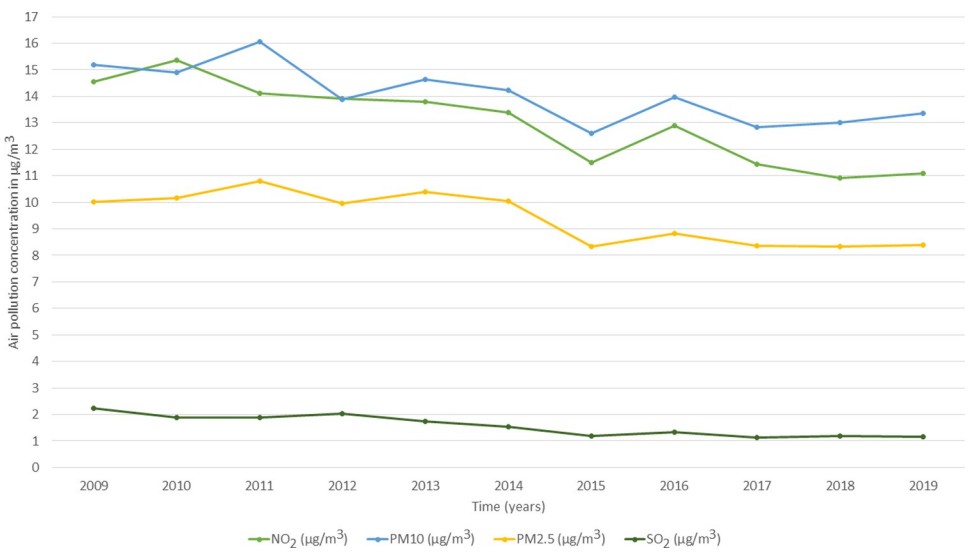

**Fig 5. The annual mean of NO$_2$, SO$_2$, PM10, and PM2.5 air pollutants at the local authority level in the UK from the year of 2009 to 2019 (N = 391 local authorities).**

### 3.3. Description of individuals' reported mental well-being

The mean score for mental well-being GHQ12 (0–12) scale was 1.8 (SD = 0.2) with 30% of responses having a GHQ12 score of 2 or more and 19% having a GHQ12 score of 4 or more. The mean score for GHQ12 (0–36) Likert scale was 11.14 (SD = 0.05) with 36% of responses having a score of 12 or more. The ICC was 0.42 and 0.49 for GHQ12 (0–12) and GHQ12 (0–36), respectively, indicating a moderate homogeneity in the well-being responses within the individual clusters over time, whilst low homogeneity (ICC = 0.16 and ICC = 0.18) was detected within the household clusters (Table 5).

### 3.4. The effect of air pollution on individuals' mental well-being

#### 3.4.1. The effect of air pollution on individuals' mental well-being at the LSOAs level.
Higher odds of poor mental well-being were observed with every 10 $\mu g/m^3$ increase in $NO_2$ (ORs ranging between 1.12–1.19), $SO_2$ (ORs ranging between 1.29–1.49), PM10 (ORs ranging between 1.19–1.34), and PM2.5 (ORs ranging between 1.30–1.53) air pollutants (Table 6).

We noticed similar results of higher odds of poor mental well-being with every 10 $\mu g/m^3$ increase in $NO_2$, PM10, and PM2.5 pollutants in bi-pollutant models adjusted for $SO_2$ (Table 7).

**Table 6. The association of individuals' mental well-being with each of $NO_2$, $SO_2$, PM10, and PM2.5 air pollutants linked at the LSOAs level in separate models (N = 349,748 surveys from 60,146 individuals).**

| | Mental well-being (GHQ0-36[a] $\geq$ 12) | | Mental well-being (GHQ0-12[b] $\geq$ 2) | | Mental well-being (GHQ0-12[b] $\geq$ 4) | |
|---|---|---|---|---|---|---|
| | Model 1 | Model 2 | Model 1 | Model 2 | Model 1 | Model 2 |
| | OR [95%CI] | OR [95%CI] | OR [95%CI] | OR [95%CI] | OR [95%CI] | OR [95%CI] |
| **Overall pollution effect** | | | | | | |
| $NO_2$ ($\mu g/m^3$) | 1.13 [1.10, 1.16]** | 1.12 [1.09, 1.15]** | 1.19 [1.16, 1.21]** | 1.14 [1.11, 1.17]** | 1.16 [1.13, 1.20]** | 1.12 [1.09, 1.16]** |
| $SO_2$ ($\mu g/m^3$) | 1.45 [1.31, 1.61]** | 1.30 [1.18, 1.44]** | 1.44 [1.30, 1.59]** | 1.29 [1.17, 1.42]** | 1.49 [1.33, 1.67]** | 1.31 [1.17, 1.47]** |
| PM10 ($\mu g/m^3$) | 1.19 [1.12, 1.27]** | 1.22 [1.15, 1.30]** | 1.34 [1.26, 1.42]** | 1.28 [1.20, 1.36]** | 1.25 [1.17, 1.34]** | 1.23 [1.15, 1.31]** |
| PM2.5 ($\mu g/m^3$) | 1.30 [1.20, 1.42]** | 1.35 [1.24, 1.47]** | 1.53 [1.41, 1.66]** | 1.44 [1.33, 1.56]** | 1.41 [1.29, 1.55]** | 1.38 [1.25, 1.51]** |
| **Between pollution effect** | | | | | | |
| $NO_2$ ($\mu g/m^3$) | 1.12 [1.09, 1.15]** | 1.11 [1.08, 1.15]** | 1.18 [1.15, 1.22]** | 1.13 [1.10, 1.17]** | 1.16 [1.12, 1.19]** | 1.12 [1.08, 1.15]** |
| $SO_2$ ($\mu g/m^3$) | 3.78 [2.98, 4.79]** | 2.21 [1.77, 2.76]** | 2.58 [2.07, 3.23]** | 1.59 [1.29, 1.96]** | 3.51 [2.73, 4.50]** | 1.94 [1.53, 2.45]** |
| PM10 ($\mu g/m^3$) | 1.17 [1.09, 1.25]** | 1.21 [1.13, 1.30]** | 1.34 [1.26, 1.43]** | 1.28 [1.19, 1.36]** | 1.24 [1.15, 1.33]** | 1.21 [1.13, 1.31]** |
| PM2.5 ($\mu g/m^3$) | 1.29 [1.17, 1.42]** | 1.36 [1.23, 1.50]** | 1.57 [1.43, 1.72]** | 1.47 [1.34, 1.61]** | 1.40 [1.26, 1.55]** | 1.36 [1.23, 1.51]** |
| **Within pollution effect** | | | | | | |
| $NO_2$ ($\mu g/m^3$) | 1.06 [0.95, 1.18] | 1.01 [0.91, 1.13] | 1.08 [0.97, 1.20] | 1.04 [0.94, 1.16] | 1.10 [0.97, 1.25] | 1.06 [0.94, 1.20] |
| $SO_2$ ($\mu g/m^3$) | 0.93 [0.78, 1.12] | 0.95 [0.80, 1.13] | 0.99 [0.83, 1.18] | 1.04 [0.87, 1.23] | 0.93 [0.76, 1.14] | 0.99 [0.81, 1.20] |
| PM10 ($\mu g/m^3$) | 1.15 [0.96, 1.38] | 1.08 [0.91, 1.29] | 1.19 [0.99, 1.42] | 1.10 [0.92, 1.31] | 1.33 [1.08, 1.64]** | 1.23 [0.99, 1.51] |
| PM2.5 ($\mu g/m^3$) | 1.17 [0.95, 1.45] | 1.09 [0.88, 1.35] | 1.26 [1.02, 1.55]* | 1.14 [0.92, 1.41] | 1.35 [1.06, 1.73]* | 1.24 [0.97, 1.59] |

**P-value <0.01

*P-value<0.05

ORs and 95%CIs are expressed in terms of 10 $\mu g/m^3$ increase in the air pollutants.

GHQ0-36[a]: GHQ scale composed of 12 questions, each scored using a Likert format: 0-1-2-3 and summed up by adding all the items generating a scale ranging from 0 to 36; the 0–36 scale is dichotomised using a cut-off score of 12 based on relevant literature into good mental well-being (score<12) and poor mental well-being (score $\geq$ 12)

GHQ0-12[b]: GHQ scale composed of 12 questions, each scored using a simple binary format: 0-0-1-1 and summed up by adding all the items generating a scale ranging from 0 to 12; the 0–12 scale is dichotomised using two cut-off scores of 2 and 4 based on relevant literature into good mental well-being (score<2 or score<4) and poor mental well-being (score $\geq$ 2 or score $\geq$ 4)

Model 1 is adjusted for age, gender and year dummies (2009–2019); Model 2 is adjusted for age, gender, ethnicity, country of birth, marital status, education, occupation, perceived financial situation, smoking status and year dummies (2009–2019).

**Table 7. The association of individuals' mental well-being with each of NO₂, PM10, and PM2.5 air pollutants linked at the LSOAs level in bi-pollutant models adjusted for SO₂ pollutant (N = 349,748 surveys from 60,146 individuals).**

| | | Mental well-being (GHQ0-36[a] $\geq$ 12) | Mental well-being (GHQ0-12[b] $\geq$ 2) | Mental well-being (GHQ0-12[b] $\geq$ 4) |
|---|---|---|---|---|
| | | OR [95%CI] | OR [95%CI] | OR [95%CI] |
| **Overall pollution effect** | | | | |
| **NO₂_SO₂ Model** | NO₂ ($\mu g/m^3$) | 1.11 [1.07, 1.14]** | 1.13 [1.10, 1.16]** | 1.11 [1.08, 1.15]** |
| | SO₂ ($\mu g/m^3$) | 1.18 [1.06, 1.31]** | 1.14 [1.02, 1.26]* | 1.17 [1.04, 1.31]** |
| **PM10_SO₂ Model** | PM10 ($\mu g/m^3$) | 1.19 [1.12, 1.27]** | 1.25 [1.18, 1.33]** | 1.20 [1.12, 1.29]** |
| | SO₂ ($\mu g/m^3$) | 1.24 [1.12, 1.37]** | 1.21 [1.09, 1.33]** | 1.24 [1.11, 1.39]** |
| **PM2.5_SO₂ Model** | PM2.5 ($\mu g/m^3$) | 1.31 [1.20, 1.43]** | 1.40 [1.29, 1.52]** | 1.34 [1.22, 1.47]** |
| | SO₂ ($\mu g/m^3$) | 1.23 [1.11, 1.37]** | 1.20 [1.08, 1.33]** | 1.23 [1.10, 1.38]** |

**P-value <0.01

*P-value<0.05

ORs and 95%CIs are expressed in terms of 10 $\mu g/m^3$ increase in the air pollutants.

GHQ0-36[a]: GHQ scale composed of 12 questions, each scored using a Likert format: 0-1-2-3 and summed up by adding all the items generating a scale ranging from 0 to 36; the 0–36 scale is dichotomised using a cut-off score of 12 based on relevant literature into good mental well-being (score<12) and poor mental well-being (score $\geq$ 12)

GHQ0-12[b]: GHQ scale composed of 12 questions, each scored using a simple binary format: 0-0-1-1 and summed up by adding all the items generating a scale ranging from 0 to 12; the 0–12 scale is dichotomised using two cut-off scores of 2 and 4 based on relevant literature into good mental well-being (score<2 or score<4) and poor mental well-being (score $\geq$ 2 or score $\geq$ 4)

Models are additionally adjusted for age, gender, ethnicity, country of birth, marital status, education, occupation, perceived financial situation, smoking status and year dummies (2009–2019).

Decomposing the overall effect of air pollution on mental well-being into *between* (spatial: across LSOAs) and *within* (temporal: across years within each LSOA) effects, revealed significant associations with poor mental well-being for the *between* effects for all the four pollutants; while no significant associations were noted for the *within* effects despite the sign of the odds ratios being largely as expected. An exception was PM10 and PM2.5 pollutants which showed significant *within* effects on poor mental well-being (GHQ0-12) only in model 1; yet these significant effects disappeared after controlling for the sociodemographic and lifestyle covariates (Table 6). Therefore, living in more polluted LSOAs was the driving cause for poor mental well-being (*between*) rather than the variation in air pollution across time *within* each LSOA.

In a sensitivity analysis, results remained the same for the *overall* and for the *between-within* effects of the four pollutants on individuals' mental well-being for individuals starting at wave 1 of the UKHLS survey (Table 2 in S1 File).

**3.4.2. The effect of air pollution on individuals' mental well-being at the local authority level.** Similar to the air pollution results at the LSOAs geographical scale, higher odds of poor mental well-being were also observed with every 10 $\mu g/m^3$ increase in NO₂ (ORs ranging between 1.10–1.16), SO₂ (ORs ranging between 1.27–1.51), PM10 (ORs ranging between 1.13–1.22), and PM2.5 (ORs ranging between 1.21–1.36) pollutants at the local authority level (Table 8).

Higher odds of poor mental well-being were also observed at the local authority level with every 10 $\mu g/m^3$ increase in NO₂, PM10, and PM2.5 pollutants in bi-pollutant models adjusted for SO₂ (Table 9).

The *between-within* (spatial-temporal) analysis of the effect of air pollution on mental well-being at the local authority level revealed similar results to the *between-within* analysis at the LSOAs level (Table 8). Thus, living in more polluted local authorities was also the driving

**Table 8. The association of individuals' mental well-being with each of NO$_2$, SO$_2$, PM10, and PM2.5 air pollutants linked at the local authority level in separate models (N = 349,748 surveys from 60,146 individuals).**

| | Mental well-being (GHQ0-36[a] ≥ 12) | | Mental well-being (GHQ0-12[b] ≥ 2) | | Mental well-being (GHQ0-12[b] ≥ 4) | |
|---|---|---|---|---|---|---|
| | Model 1 | Model 2 | Model 1 | Model 2 | Model 1 | Model 2 |
| | OR [95%CI] | OR [95%CI] | OR [95%CI] | OR [95%CI] | OR [95%CI] | OR [95%CI] |
| **Overall pollution effect** | | | | | | |
| NO$_2$ (µg/m$^3$) | 1.11 [1.07, 1.15]** | 1.10 [1.07, 1.14]** | 1.16 [1.12, 1.20]** | 1.13 [1.09, 1.16]** | 1.13 [1.08, 1.17]** | 1.10 [1.06, 1.14]** |
| SO$_2$ (µg/m$^3$) | 1.51 [1.27, 1.80]** | 1.44 [1.22, 1.71]** | 1.42 [1.20, 1.68]** | 1.34 [1.14, 1.58]** | 1.36 [1.12, 1.65]** | 1.27 [1.05, 1.53]** |
| PM10 (µg/m$^3$) | 1.13 [1.04, 1.23]** | 1.17 [1.09, 1.26]** | 1.22 [1.13, 1.32]** | 1.21 [1.13, 1.30]** | 1.15 [1.06, 1.26]** | 1.17 [1.08, 1.26]** |
| PM2.5 (µg/m$^3$) | 1.21 [1.08, 1.35]** | 1.27 [1.15, 1.41]** | 1.36 [1.23, 1.51]** | 1.35 [1.23, 1.49]** | 1.25 [1.12, 1.41]** | 1.28 [1.15, 1.43]** |
| **Between pollution effect** | | | | | | |
| NO$_2$ (µg/m$^3$) | 1.09 [1.04, 1.13]** | 1.09 [1.05, 1.13]** | 1.14 [1.10, 1.19]** | 1.11 [1.07, 1.15]** | 1.11 [1.07, 1.16]** | 1.09 [1.05, 1.13]** |
| SO$_2$ (µg/m$^3$) | 3.72 [2.53, 5.46]** | 2.62 [1.87, 3.67]** | 3.03 [2.08, 4.41]** | 2.11 [1.50, 2.96]** | 3.58 [2.40, 5.35]** | 2.39 [1.67, 3.43]** |
| PM10 (µg/m$^3$) | 1.09 [0.99, 1.20] | 1.16 [1.06, 1.26]** | 1.23 [1.12, 1.34]** | 1.22 [1.13, 1.33]** | 1.13 [1.03, 1.24]* | 1.16 [1.06, 1.26]** |
| PM2.5 (µg/m$^3$) | 1.16 [1.02, 1.33]* | 1.26 [1.12, 1.42]** | 1.39 [1.23, 1.58]** | 1.38 [1.24, 1.54]** | 1.23 [1.07, 1.41]** | 1.27 [1.13, 1.44]** |
| **Within pollution effect** | | | | | | |
| NO$_2$ (µg/m$^3$) | 1.07 [0.94, 1.22] | 1.06 [0.93, 1.21] | 1.14 [1.00, 1.30] | 1.12 [0.99, 1.28] | 1.08 [0.93, 1.26] | 1.07 [0.92, 1.25] |
| SO$_2$ (µg/m$^3$) | 1.08 [0.82, 1.44] | 1.01 [0.77, 1.35] | 1.13 [0.85, 1.49] | 1.08 [0.81, 1.43] | 0.95 [0.69, 1.32] | 0.90 [0.65, 1.25] |
| PM10 (µg/m$^3$) | 1.11 [0.91, 1.35] | 1.09 [0.89, 1.32] | 1.05 [0.86, 1.28] | 1.02 [0.84, 1.24] | 1.22 [0.97, 1.54] | 1.19 [0.94, 1.49] |
| PM2.5 (µg/m$^3$) | 1.09 [0.86, 1.39] | 1.06 [0.83, 1.34] | 1.09 [0.86, 1.39] | 1.04 [0.82, 1.32] | 1.17 [0.88, 1.55] | 1.12 [0.85, 1.49] |

**P-value <0.01

*P-value<0.05

ORs and 95%CIs are expressed in terms of 10 µg/m$^3$ increase in the air pollutants.

GHQ0-36[a]: GHQ scale composed of 12 questions, each scored using a Likert format: 0-1-2-3 and summed up by adding all the items generating a scale ranging from 0 to 36; the 0–36 scale is dichotomised using a cut-off score of 12 based on relevant literature into good mental well-being (score<12) and poor mental well-being (score ≥ 12)

GHQ0-12[b]: GHQ scale composed of 12 questions, each scored using a simple binary format: 0-0-1-1 and summed up by adding all the items generating a scale ranging from 0 to 12; the 0–12 scale is dichotomised using two cut-off scores of 2 and 4 based on relevant literature into good mental well-being (score<2 or score<4) and poor mental well-being (score ≥ 2 or score ≥ 4)

Model 1 is adjusted for age, gender and year dummies (2009–2019); Model 2 is adjusted for age, gender, ethnicity, country of birth, marital status, education, occupation, perceived financial situation, smoking status and year dummies (2009–2019).

cause for poor mental well-being (*between*) rather than the variation in air pollution across time *within* each local authority.

In a sensitivity analysis for individuals starting at wave 1 of the UKHLS survey, results remained unchanged for the *overall* and for the *between-within* effects of the four pollutants on individuals' mental well-being at the local authority level (Table 3 in S1 File).

### 3.5. The association of air pollution with individuals' mental well-being by ethnicity and country of birth

**3.5.1. The association of air pollution at the LSOAs level with individuals' mental well-being by ethnicity and country of birth.** At the LSOAs level, Pakistani/Bangladeshi, other-white, and other ethnicities group as well as non-UK born individuals showed higher odds of poor mental well-being compared to British-white and UK-born individuals, respectively, with every 10 µg/m$^3$ increase in SO$_2$, PM10, and PM2.5 pollutants only with the GHQ0-12 well-being measure (Fig 6). Nevertheless, most of these significant differences disappeared in a cohort sub-analysis for only individuals recruited at wave 1 of the UKHLS survey (Fig 1 in S1 File). In addition, we observed no significant ethnic differences for the *between* or *within*

**Table 9. The association of individuals' mental well-being with each of NO$_2$, PM10, and PM2.5 air pollutants linked at the local authority level in bi-pollutant models adjusted for SO$_2$ pollutant (N = 349,748 surveys from 60,146 individuals).**

| | | Mental well-being (GHQ0-36[a] $\geq$ 12) | Mental well-being (GHQ0-12[b] $\geq$ 2) | Mental well-being (GHQ0-12[b] $\geq$ 4) |
| --- | --- | --- | --- | --- |
| | | OR [95%CI] | OR [95%CI] | OR [95%CI] |
| **Overall pollution effect** | | | | |
| **NO$_2$_SO$_2$ Model** | NO$_2$ (µg/m$^3$) | 1.09 [1.05, 1.12]** | 1.12 [1.08, 1.16]** | 1.10 [1.06, 1.14]** |
| | SO$_2$ (µg/m$^3$) | 1.25 [1.05, 1.49]* | 1.10 [0.93, 1.31] | 1.07 [0.88, 1.30] |
| **PM10_SO$_2$ Model** | PM10 (µg/m$^3$) | 1.13 [1.05, 1.22]** | 1.19 [1.10, 1.28]** | 1.15 [1.06, 1.24]** |
| | SO$_2$ (µg/m$^3$) | 1.36 [1.14, 1.61]** | 1.23 [1.04, 1.45]* | 1.18 [0.97, 1.43] |
| **PM2.5_SO$_2$ Model** | PM2.5 (µg/m$^3$) | 1.22 [1.10, 1.35]** | 1.32 [1.19, 1.46]** | 1.25 [1.12, 1.40]** |
| | SO$_2$ (µg/m$^3$) | 1.34 [1.13, 1.59]** | 1.21 [1.02, 1.43]* | 1.16 [0.96, 1.40] |

**P-value <0.01

*P-value<0.05

ORs and 95%CIs are expressed in terms of 10 µg/m$^3$ increase in the air pollutants.

GHQ0-36[a]: GHQ scale composed of 12 questions, each scored using a Likert format: 0-1-2-3 and summed up by adding all the items generating a scale ranging from 0 to 36; the 0–36 scale is dichotomised using a cut-off score of 12 based on relevant literature into good mental well-being (score<12) and poor mental well-being (score $\geq$ 12)

GHQ0-12[b]: GHQ scale composed of 12 questions, each scored using a simple binary format: 0-0-1-1 and summed up by adding all the items generating a scale ranging from 0 to 12; the 0–12 scale is dichotomised using two cut-off scores of 2 and 4 based on relevant literature into good mental well-being (score<2 or score<4) and poor mental well-being (score $\geq$ 2 or score $\geq$ 4)

Models are additionally adjusted for age, gender, ethnicity, country of birth, marital status, education, occupation, perceived financial situation, smoking status and year dummies (2009–2019).

effects of air pollution at the LSOAs level on mental well-being with exception of *between* effects for PM10 and PM2.5 acting as protective factors (ORs<1) against poor mental well-being for people from Indian, Black/African/Caribbean and mixed ethnicities origin (Figs 3 and 4 in S1 File).

It is worth to note that examining the association between ethnicity and mental well-being revealed higher odds of poor mental well-being among people from Pakistani/Bangladeshi (GHQ0-36 $\geq$ 12: OR = 1.17, 95%CI = 1.06–1.29; GHQ0-12 $\geq$ 2: OR = 1.09, 95%CI = 1.00–1.19; GHQ0-12 $\geq$ 4: OR = 1.13, 95%CI = 1.02–1.25) and mixed ethnicities (GHQ0-36 $\geq$ 12: OR = 1.15, 95%CI = 1.01–1.32; GHQ0-12 $\geq$ 2: OR = 1.21, 95%CI = 1.08–1.37; GHQ0-12 $\geq$ 4: OR = 1.26, 95%CI = 1.10–1.43) origin in comparison to the British-white, even after adjusting for socio-demographics and lifestyle covariates (Table 1 in S1 File).

**3.5.2. The association of air pollution at the local authority level with individuals' mental well-being by ethnicity and country of birth.** Similar to the LSOAs level, analysis at the local authority level showed higher odds of poor mental well-being with increasing concentrations of SO$_2$, PM10, and PM2.5 pollutants among people from Pakistani/Bangladeshi and other ethnicities origin compared to British-white and among non-UK born people compared to natives. People from an Indian origin also showed higher odds of poor mental well-being than the British-white with every 10 µg/m$^3$ increase in SO$_2$ pollution at the local authority level (Fig 7). Yet, these significant differences disappeared in a cohort sub-analysis for only individuals recruited at wave 1 of the UKHLS survey (Fig 2 in S1 File). Similarly, no significant ethnic differences for the *between* or *within* effects of air pollution at the local authority level on mental well-being were observed with exception of *between* effects for PM10 and PM2.5 acting as protective factors (ORs<1) against poor mental well-being for people from Indian, Black/African/Caribbean and mixed ethnicities origin (Figs 5 and 6 in S1 File).

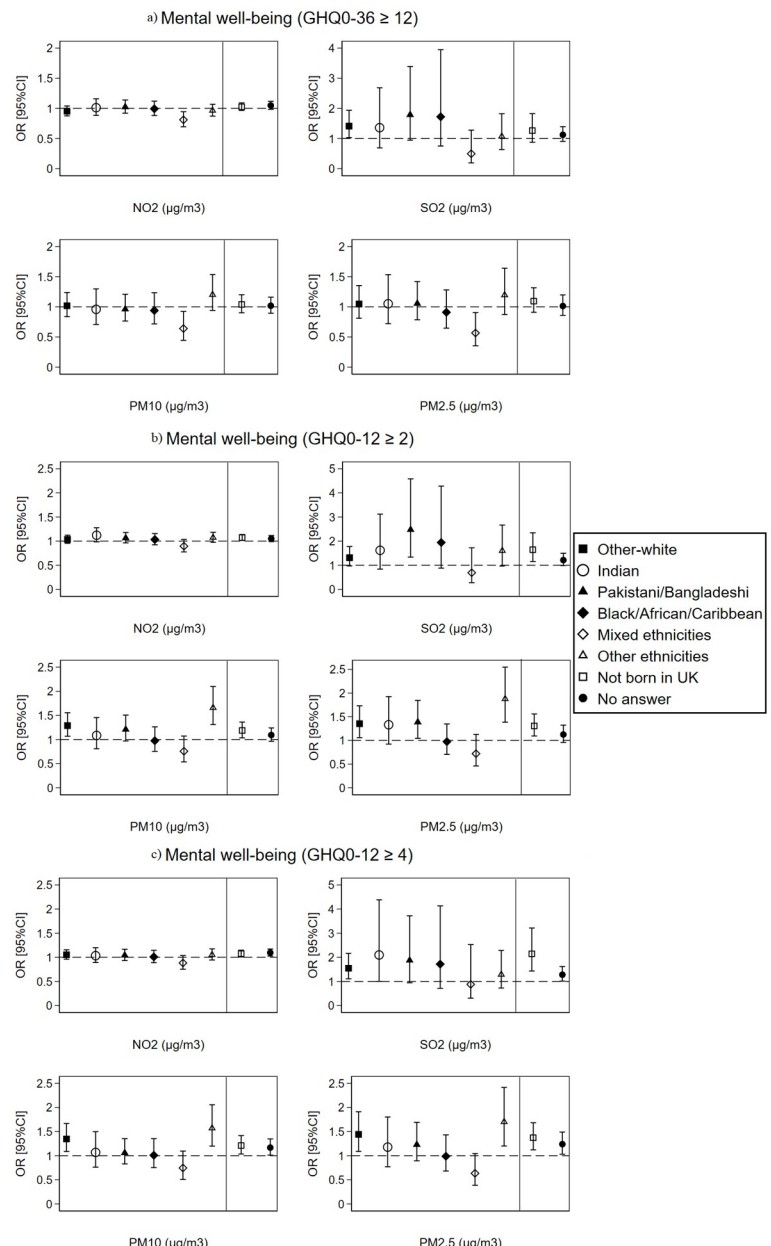

**Fig 6. The overall effect of air pollution linked at the LSOAs level on individuals' mental well-being by ethnicity and country of birth (N = 349,748 surveys from 60,146 individuals).** The dashed line is placed at OR = 1 as a cut-off for statistically insignificant results; The solid line separates between the air pollution-ethnicity interaction models and the air pollution-country of birth interaction models; Air pollution-ethnicity interaction models are adjusted for country of birth, age, gender, marital status, education, occupation, perceived financial situation, smoking status, and year dummies (2009 to 2019); Air pollution-country of birth interaction models are adjusted for ethnicity, age, gender, marital status, education, occupation, perceived financial situation, smoking status, and year dummies (2009 to 2019).

## 4. Discussion

This study showed the negative impact of $NO_2$, $SO_2$, PM10 and PM2.5 air pollution (linked at two geographical scales: coarse local authorities and detailed LSOAs) on mental well-being in the UK for individuals followed from the year 2009 up to 2019. These results are supported by

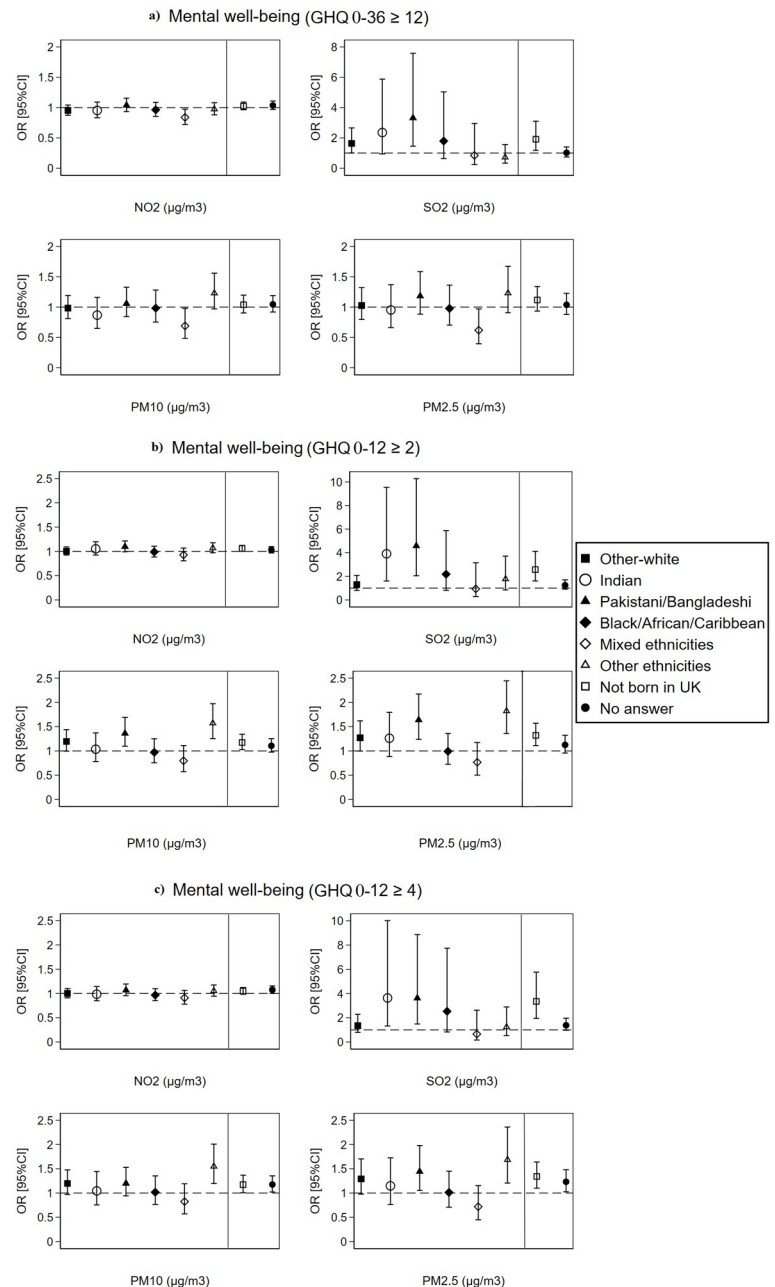

**Fig 7. The overall effect of air pollution linked at the local authority level on individuals' mental well-being by ethnicity and country of birth (N = 349,748 surveys from 60,146 individuals).** The dashed line is placed at OR = 1 as a cut-off for statistically insignificant results; The solid line separates between the air pollution-ethnicity interaction models and the air pollution-country of birth interaction models; Air pollution-ethnicity interaction models are adjusted for country of birth, age, gender, marital status, education, occupation, perceived financial situation, smoking status, and year dummies (2009 to 2019); Air pollution-country of birth interaction models are adjusted for ethnicity, age, gender, marital status, education, occupation, perceived financial situation, smoking status, and year dummies (2009 to 2019).

relevant literature whereby exposure to ambient air pollution has been shown to affect negatively individuals' mental well-being and contribute to increased rates of mental health problems such as autism spectrum disorders [13], schizophrenia [14], dementia [15], psychotic

experiences [16, 17], cognitive disabilities [18], anxiety and major depressive disorders [19]. Evidence from China revealed an elevation in the rate of depressive symptoms and poorer self-reported mental well-being with long-term exposure to air pollution [65–67]. Similarly, exposure to $NO_2$ and particulate matter (PM10 and PM2.5) air pollution in the Netherlands was positively associated with poor mental health and prescription of anti-anxiety drugs [68]. In South Korea, increased exposure to $NO_2$, $SO_2$, and PM10 pollutants resulted in higher hazards for suicide death [69]. In addition, a recent systematic literature review and meta-analysis of 22 articles revealed that exposure to PM2.5 pollution increases the risk for depression and anxiety with a pooled odd ratio estimate of 1.10 for every 10 μg/m$^3$ increase in PM2.5 concentration [70].

The observed positive association between air pollution and poor mental well-being in this study can be explained by four factors. The first explanation is through the biological mechanisms of air pollutants on the human central nervous system and neuro-behavioural processes [1, 5, 9, 10]. Air pollution particles of small diameters such as PM2.5 are capable of initiating oxidative stress and forming inflammatory cytokines that infiltrate the blood-brain barrier causing neurodegeneration and neuroinflammation [11]. The second explanation is through the aesthetic and odorous nuisance caused by air pollution, which results in avoidance behaviour and inhibition of psychological-supporting outdoor activities and sports. This in turn leads to reduced happiness and life satisfaction and to elevated levels of stress, anxiety, loneliness, and poor mental well-being [2, 23–25]. The third explanation is related to experiential anxiety and worrying feelings about one's physical health and future [27]. Through substantive research, people are made aware of the negative impact of air pollution on human's physical health and the higher risk for acute and chronic diseases such as cardiovascular, respiratory, cancer and immune system diseases [8, 9, 28]. People living in highly polluted areas might experience stress and anxiety about physical illness, which is reflected in poorer mental well-being. The last explanation is manifested in the indirect effect of air pollution on mental well-being through the physical health of individuals. People who already suffer from physical illness often also suffer from poor mental well-being. Based on an international study that utilises data from the World Health Organisation, 9% to 23% of patients with one or more chronic physical health conditions displayed symptoms of depression [71]. Thus, air pollution might be associated with poor mental well-being because individuals are also suffering from a physical health condition. Conducting a simple t-test on the UKHLS data showed a higher GHQ12 (0–12) mean score (mean = 2.44, SD = 3.46) among individuals with at least one physical health condition (e.g., asthma, arthritis, coronary heart disease, cancer, liver illness, chronic bronchitis, diabetes, blood pressure) in comparison to a GHQ12(0–12) mean score of 1.68 (SD = 2.92) for those with non-reported physical health condition (t-test P-value = 0.000).

Despite the existence of literature on the topic of air pollution and poor mental well-being, this study went a further step in analysing the *spatial-temporal* effects of air pollution on mental well-being using a *between-within* longitudinal design. Additionally, we carried out the analysis at two geographical scales, coarse local authorities and detailed LSOAs, which forms another novelty of the present study. The *between-within* analysis is extensively used in the fields of economics, behavioural finance, and strategic management [29], yet little used in health research [72]; and no previous study has examined the *between-within* effects of air pollution on mental well-being. Our study revealed significant *between* effects for $NO_2$, $SO_2$, PM10 and PM2.5 pollutants on poor mental well-being at both the LSOAs and the local authority geographical scales, while no significant *within* effects were noted. Thus, individuals residing in LSOAs or local authorities with higher average concentrations of the four pollutants across the 11 years of follow up exhibited poorer mental well-being than individuals residing in LSOAs or local authorities with lower pollution concentrations. This shows the

importance of the spatial dimension in the association between air pollution and mental well-being whether at the coarse local authorities or at the detailed LSOAs geographical level given that both geographical scales resulted in similar findings. Nevertheless, the non-significant *within* (temporal) effects could be related to the low variation of yearly air pollution concentrations across the 11 years of follow up, particularly for $SO_2$ pollutant as shown in Fig 4 for LSOAs and in Fig 5 for local authorities. Therefore, further research with longer follow up time is needed to allow for more temporal variation in air pollution which might result in significant *within* effects.

In the second part of this study, we attempted to examine the moderating effect of ethnicity and country of birth on the association between air pollution and individuals' mental well-being. We hypothesised that the lower socio-economic status of ethnic minorities and living in more disadvantaged neighbourhoods, near major roads and transportation networks can result in higher levels of stress, anxiety, and mental health problems among ethnic minority groups; thus, moderating the association between air pollution and mental well-being. However, our findings did not reveal much difference in the overall and in the *between* and *within* effects of air pollution on mental well-being across the ethnic groups; with exception for Pakistani/Bangladeshi, other-white (only at the LSOAs level but not at the local authority level), Indians (only for $SO_2$ pollution at the local authority level), other ethnicities, and non-UK born individuals who showed higher odds of poor mental well-being than the British-White and UK-born individuals with increasing concentrations of $SO_2$, PM10, and PM2.5 pollutants. The poorer mental well-being observed among the Pakistani/Bangladeshi, other ethnicities group, and non-UK born individuals could be related to the socio-economic and lifestyle differences or to place-related contextual differences. Literature on ethnic inequalities in health has shown that ethnic minorities often live in more disadvantaged communities, and have lower socio-economic status, lower healthcare coverage and higher job/income insecurity [34, 73, 74], which increases their risk of physical and mental illness. Nevertheless, our analysis adjusted for the main socio-economic and lifestyle factors of individuals including age, gender, marital status, education, occupation, financial situation, and cigarette smoking. Therefore, the most likely explanation for the more pronounced effect of $SO_2$, PM10, and PM2.5 pollution on poor mental well-being among Pakistani/Bangladeshi, other ethnicities, and non-UK born individuals could be related to the contextual factors and place of residence. Ethnic minorities and immigrants often choose to reside in large cities and highly urbanised regions, near major roads and key transportation networks to simplify their commuting and working conditions [75]. In addition, ethnic minorities often live in low-priced social housing offered by local authorities, which is often situated in more deprived ethnic concentration neighbourhoods [35]. These place-related factors can result in more pronounced effect of air pollution on mental well-being among ethnic minorities and immigrants due to greater exposure to air pollution resulting from vehicles, factories, and burning of fossil fuels. In a supplementary analysis, we show through Chi2 square tabulation that a very high percentage of non-UK born individuals (93%) and of ethnic minorities including Pakistani/Bangladeshi (99%) reside in urban areas; whereas this percentage is much lower for British-white (71%) and UK-born (75%) individuals (Table 4 in S1 File). It should be noted, however, that decomposing the overall effect of air pollution into *between* (spatial) and *within* (temporal) effects at both the LSOAs and local authority levels did not show significant differences among the ethnic groups, not even for the Pakistani/Bangladeshi and non-UK born individuals. Thus, we cannot be conclusive that residing in more polluted areas is the key explanation for the more pronounced effect of $SO_2$, PM10, and PM2.5 pollution on mental well-being among Pakistani/Bangladeshi and non-UK born individuals. Furthermore, in a cohort sub-analysis for individuals recruited at wave 1 of the UKHLS survey (Figs 1 and 2 in S1 File), the differences noted with respect to the

Pakistani/Bangladeshi, other ethnicities group, and non-UK born individuals disappeared. Therefore, our study shows no conclusive evidence of ethnicity or nativity differences in the association between air pollution and poor mental well-being.

Despite the new insights provided by this study, it is important to discuss its limitations. The first limitation is related to the design of the study in which individual-level data from the UKHLS survey was linked to yearly air pollution contextual data at the local authority level. Therefore, individuals residing within a respective local authority and a respective year were assigned to the same value of air pollution exposure. However, we also linked the air pollution data to the UKHLS at the LSOAs level (the lowest available geography level at the UKHLS due to ethical considerations) which minimised the exposure bias. Air pollution linked at both, the local authority and the LSOAs level, revealed similar results. For future research, we recommended the usage of data sources that allow linkages of air pollution at the postcode level, the lowest available geography in the UK. Second, our analysis included all individuals recruited at different waves of the UKHLS survey, that had at least two observations through the follow up time (2009–2019). Hence, the time window for follow up differed between the study participants, some were followed for 11 years across all the 10 waves while others entered the study at later waves and were followed for a shorter period of time. It should be noted, however, that similar results were observed in a sensitivity analysis on wave 1 cohort, except for the analysis by ethnic groups whereby Pakistani/Bangladeshi, other ethnicities, and non-UK born individuals did not show significant differences in the association of air pollution with mental well-being at both the LSOAs and local authority geographical scales (Tables 2, 3 and Figs 1 and 2 in S1 File). This can be explained by the UKHLS sample design which involved ethnic minority boost samples at waves 1 and 6 of data collection to enable ethnicity-focused research. Therefore, by conducting analysis on only wave 1 cohort, we are missing the second ethnic minority boost sample at wave 6 which resulted in the observed differences. Finally, the UKHLS survey included longitudinal weights that adjust for the overrepresentation of some groups, such as the ethnic minority groups; thus, allowing for greater generalisation of the estimates. However, we could not add the longitudinal weights into our analysis as this requires that all individuals be followed until the last wave (wave 10) of the survey, which was not the case.

## 5. Conclusion

Using a longitudinal panel design that involves linking individual to context-level data at two geographical scales (coarse local authorities and detailed LSOAs) and a *between-within* analysis, this study highlights the negative effect of air pollution on individuals' mental well-being over space and time and emphasises the importance of the spatial dimension in the shaping of this association. Thus, environmental policies to reduce air pollution emissions with a core of spatial planning can eventually improve the mental well-being of people residing in the UK. There is, however, less conclusive evidence on the moderating effect of ethnicity.

## Supporting information

**S1 File. Additional analysis and sensitivity check-ups.**
(DOCX)

## Author Contributions

**Conceptualization:** Mary Abed Al Ahad, Urška Demšar, Frank Sullivan, Hill Kulu.

**Data curation:** Mary Abed Al Ahad.

**Formal analysis:** Mary Abed Al Ahad.

**Funding acquisition:** Urška Demšar, Frank Sullivan, Hill Kulu.

**Investigation:** Mary Abed Al Ahad.

**Methodology:** Mary Abed Al Ahad.

**Project administration:** Mary Abed Al Ahad.

**Resources:** Mary Abed Al Ahad.

**Software:** Mary Abed Al Ahad.

**Supervision:** Urška Demšar, Frank Sullivan, Hill Kulu.

**Validation:** Mary Abed Al Ahad.

**Visualization:** Mary Abed Al Ahad.

**Writing – original draft:** Mary Abed Al Ahad.

**Writing – review & editing:** Mary Abed Al Ahad, Urška Demšar, Frank Sullivan, Hill Kulu.

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
