## [Decision Letter · Decision Letter 0]

9 Dec 2021

PONE-D-21-31292Air pollution and individuals’ mental well-being in the adult population in United Kingdom: A spatial-temporal longitudinal study and the moderating effect of ethnicityPLOS ONE

Dear Dr. Abed Al Ahad,

Thank you for submitting your manuscript to PLOS ONE. After careful consideration, we feel that it has merit but does not fully meet PLOS ONE’s publication criteria as it currently stands. Therefore, we invite you to submit a revised version of the manuscript that addresses the points raised during the review process.

We look forward to receiving your revised manuscript.

Kind regards,

Bijaya Kumar Padhi, PhD, MPH

Academic Editor

PLOS ONE

a) Did participants provide their written or verbal informed consent to participate in this study?

3. You indicated that you had ethical approval for your study. In your Methods section, please ensure you have also stated whether you obtained consent from parents or guardians of the minors included in the study or whether the research ethics committee or IRB specifically waived the need for their consent.

4. Thank you for stating the following in the Funding Section of your manuscript:

“: This paper is part of a PhD project that is funded by the St Leonard’s PhD scholarship, University of St Andrews, Scotland, United Kingdom.”

“This paper is part of a PhD project that is funded by the St Leonard’s PhD scholarship, University of St Andrews, Scotland, United Kingdom.”

7. Please note that in order to use the direct billing option the corresponding author must be affiliated with the chosen institute. Please either amend your manuscript to change the affiliation or corresponding author, or email us at plosone@plos.org with a request to remove this option.

8. Your ethics statement should only appear in the Methods section of your manuscript. If your ethics statement is written in any section besides the Methods, please move it to the Methods section and delete it from any other section. Please ensure that your ethics statement is included in your manuscript, as the ethics statement entered into the online submission form will not be published alongside your manuscript.

9. We note that Figure 3 in your submission contain [map/satellite] images which may be copyrighted. All PLOS content is published under the Creative Commons Attribution License (CC BY 4.0), which means that the manuscript, images, and Supporting Information files will be freely available online, and any third party is permitted to access, download, copy, distribute, and use these materials in any way, even commercially, with proper attribution. For these reasons, we cannot publish previously copyrighted maps or satellite images created using proprietary data, such as Google software (Google Maps, Street View, and Earth). For more information, see our copyright guidelines: http://journals.plos.org/plosone/s/licenses-and-copyright.

 a. You may seek permission from the original copyright holder of Figure 3 to publish the content specifically under the CC BY 4.0 license. 

:

Additional Editor Comments:

We have received reviews from two reviewers for your manuscript “Air pollution and individuals’ mental well-being in the adult population in United Kingdom: A spatial-temporal longitudinal study and the moderating effect of ethnicity”. At this time, the manuscript will require substantial revision before it can be considered for publication. The reviewers recommended that you make substantial amendments to your manuscript. Please respond within the next 14 days to all comments raised by the reviewers. You can also submit a revised version of your manuscript at that time. We encourage you to submit your documents with tracked changes to highlight the revisions.

Reviewers' comments:

Reviewer's Responses to Questions

**Comments to the Author**

1. Is the manuscript technically sound, and do the data support the conclusions?

Reviewer #1: Yes

Reviewer #2: Yes

2. Has the statistical analysis been performed appropriately and rigorously? 

Reviewer #1: Yes

Reviewer #2: No

3. Have the authors made all data underlying the findings in their manuscript fully available?

Reviewer #1: No

Reviewer #2: Yes

4. Is the manuscript presented in an intelligible fashion and written in standard English?

Reviewer #1: No

Reviewer #2: Yes

5. Review Comments to the Author

Reviewer #1: This study is interesting because the authors utilized the datasets from a large cohort to investigate the associations between mental health outcomes and exposures to NO2, SO2, PM2.5, and PM10. The study is properly designed, and data analyses are adequate with new findings. The data interpretations and discussions are adequate.

Specific comments:

Line 99: Please break this sentence into two separately focusing impacts of PM1 and PM2.5. A part of PM2.5 cannot cross blood brain barriers.

Fig 4 title: correct subscripts.

Line 209: Please provide a reference justifying that 1x1 km spatial resolution is sufficient for selected air pollutants for this type of association studies. Please add more information on this.

Lines 224 – 234: Besides citing references please also add some information stating the connections between these factors and targeted pollutants and health outcomes.

Table 5: correct subscripts for air pollutants.

Reviewer #2: The authors tried to examine the effects of ambient air pollution on mental health by measuring mental well being by a suitable tool. I also agree with authors that evidence base of the association between air pollution and mental health across the globe is week. In that context this study is important and has good scientific merits. The largest advantage of the study is the large sample size with repeated measure over 11 years. Though exposure misclassification is an issue as authors highlighted as population exposure assigned to individual, but that was only available option to them. However, there are lot of methodological issues, unless that are addressed, I cannot recommend for publication. These are the following issues

1. I couldn’t understand why they dichotomized each item of GHQ12 tools. Instead, they could have added all considering each item measured in an ordinal scale of 4 points to have better precision in measurement. By doing so they reduced the true variability mechanical which led to decrease in precision of the measurement to some extent.

2. The most serious flaw of the analysis is not to report the distribution of the response of interest. As far as existing literature is concern that is supposed to be a negatively skewed distribution. Though I don’t know what their distribution was looked like, but my guess it is too both side truncated(bounded) negatively skewed distribution. Whatever means and sds reported by them in the discussion indicated presence of severe skewness. Question is how valid a Gaussian Mixed Effects Model is under this background of response distribution. No such attempt was made by the authors to check such validity of the multilevel model they used. Without that one cannot rely on their estimates or findings.

3. I recommend a Beta-Binomial mixed effects model to be used for this analysis which is commonly used by people with tool-based measurement of psychological features in the literature. Authors can convert the score into a proportion scale (0,1) by minimax normalization method. If any score attends exactly 0, or 1 do the necessary step to shrink them within (0,1). Report the results in terms OR per 10ug/m3 increase in pollutant. OR should be interpreted as the ratio of the odds of poor mental health wellbeing for every 10ug/m3 increase in ambient concentration of the pollutant. Please go through the relevant literature will guide you. As data wouldn’t be binary, the interpretation in terms of OR is a bit tricky.

4. After understanding the data structure, I feel multilevel model may not be needed. There is no reason to believe that the temporal effects will vary over geography. Both temporal and cluster effects can be assumed independent considering two independent random intercepts. Secondly, the levels should be finalized by testing the variances step by step (H0: sigma_b=0 vs H1:sigma_b!=0). If nested model, then nested variances should be tested. Look at random effects model literature. ICC based assessment may not be correct always as that ignores the nested structure.

5. I am confused with within vs between effects of pollutants. After reading their process of computation I understand they all are same measured with different level of precision. 1) manually removed the temporal variation and estimated effect size 2) manually removed spatial variation and estimated effect size 3) statistically removed spatiotemporal variation and estimated effects size. Where are the differences? First fix the questions you are asking. The possible questions could be 1) Do the effects of pollution vary over geography? 2) Do the effects vary over duration of exposure? Stratification by interaction term should be the solution of it.

6. Supplementary analysis should be done by either baseline logistic model or ordered logistic, not by binary logistic model. By multiple binary logistic models’ precision of the estimates would be compromised.

7. In the abstract authors reported regression coefficients but named as OR, I guess. As higher score means worst in mental wellbeing, one should expect a positive association with pollution, OR should be >1.

8. Line 385-389: Authors stating in text about high score but reporting regression coefficients doesn’t go with sentence. Is it the mean difference from reference ethnic group? If so, state accordingly.

9. Looking at the correlation matrix among pollutant, it is worthy to try multipollutant model with SO2, NO2 and PM2.5. Look at variance inflation factor, if less than 8/10, don’t bother, it will work. Best way to assess is add one at a time if AIC decreases you can retain. In case of multicollinearity AIC should increase after adding them in a single model.

6. PLOS authors have the option to publish the peer review history of their article (what does this mean?). If published, this will include your full peer review and any attached files.

Reviewer #1: No

Reviewer #2: No

---

## [Author Response · Author response to Decision Letter 0]

14 Jan 2022

PONE-D-21-31292

Air pollution and individuals’ mental well-being in the adult population in United Kingdom: A spatial-temporal longitudinal study and the moderating effect of ethnicity

PLOS ONE

a) Did participants provide their written or verbal informed consent to participate in this study?

Response: For this study, we didn’t need informed consent from participants because we did secondary analysis on the “Understanding Society: The UK Household Longitudinal Study (UKHLS)” dataset. The UKHLS team have already obtained written informed consent from participants.

We have added into the manuscript a section on Ethical considerations under the methods which includes the above information.

Response: Not applicable.

3. You indicated that you had ethical approval for your study. In your Methods section, please ensure you have also stated whether you obtained consent from parents or guardians of the minors included in the study or whether the research ethics committee or IRB specifically waived the need for their consent.

Response: We have added to the methods a section about Ethical considerations as follows: “This paper was granted ethical approval on the 14th of May 2020 by the authors’ affiliated institution (School of Geography and Sustainable Development Ethics Committee, acting on behalf of the University Teaching and Research Ethics Committee (UTREC) at the University of St Andrews). The paper uses secondary adult (age 16+) data from the “Understanding Society: The UK Household Longitudinal Study (UKHLS)”; thus, participants’ informed consent is not applicable. The University of Essex responsible for the UKHLS data collection and management has already obtained written informed consent from all the study participants (37). Requesting consent for health record linkage was approved at Wave 1 by the National Research Ethics Service (NRES) Oxfordshire REC A (08/H0604/124), and at Wave 4 by NRES Southampton REC A (11/SC/0274). Approval for the collection of biosocial data by trained nurses in Waves 2 and 3 of the main survey was obtained from the National Research Ethics Service (Understanding Society - UK Household Longitudinal Study: A Biosocial Component, Oxfordshire A REC, Reference: 10/H0604/2)”.

4. Thank you for stating the following in the Funding Section of your manuscript:

“: This paper is part of a PhD project that is funded by the St Leonard’s PhD scholarship, University of St Andrews, Scotland, United Kingdom.”

“This paper is part of a PhD project that is funded by the St Leonard’s PhD scholarship, University of St Andrews, Scotland, United Kingdom.”

Response: We have removed the Funding statement from the acknowledgement section of the manuscript. The funding statement in the online submission form is correct as it stands: “This paper is part of a PhD project that is funded by the St Leonard’s PhD scholarship, University of St Andrews, Scotland, United Kingdom.”

5. In your Data Availability statement, you have not specified where the minimal data set underlying the results described in your manuscript can be found. PLOS defines a study's minimal data set as the underlying data used to reach the conclusions drawn in the manuscript and any additional data required to replicate the reported study findings in their entirety. All PLOS journals require that the minimal data set be made fully available. 

Upon re-submitting your revised manuscript, please upload your study’s minimal underlying data set as either Supporting Information files or to a stable, public repository and include the relevant URLs, DOIs, or accession numbers within your revised cover letter. 

Important: If there are ethical or legal restrictions to sharing your data publicly, please explain these restrictions in detail. 

Response: We cannot make the data underlying our analysis publicly available due to ethical and legal restrictions. We are using the “Understanding Society: The UK Household Longitudinal Study (UKHLS)” dataset which is an initiative funded by the Economic and Social Research Council and various Government Departments, with scientific leadership by the Institute for Social and Economic Research, University of Essex, and survey delivery by NatCen Social Research and Kantar Public. These data are protected by a copyright license and strictly distributed by the UK Data Service which is the largest digital repository for quantitative and qualitative social science and humanities research data in the UK. Therefore, data underlying our analysis can only be accessed through the UK Data Service for authorized researchers from the following URL: https://beta.ukdataservice.ac.uk/datacatalogue/series/series?id=2000053

We have added the above explanation into the cover letter.

Response: We have made changes to our Data Availability Statement because the data cannot be made publicly available by the researchers due to ethical and legal considerations. We have added the updated data availability statement at the end of the cover letter.

7. Please note that in order to use the direct billing option the corresponding author must be affiliated with the chosen institute. Please either amend your manuscript to change the affiliation or corresponding author, or email us at plosone@plos.org with a request to remove this option.

Response: The corresponding author is affiliated with the chosen institute (University of St. Andrews). We have amended the information provided on the system and in the Title page to reflect that by adding a full stop after the St. Andrews. 

8. Your ethics statement should only appear in the Methods section of your manuscript. If your ethics statement is written in any section besides the Methods, please move it to the Methods section and delete it from any other section. Please ensure that your ethics statement is included in your manuscript, as the ethics statement entered into the online submission form will not be published alongside your manuscript.

Response: We have added the ethics statement into the Methods section of the manuscript on line 357 and removed it from the Acknowledgement section. 

9. We note that Figure 3 in your submission contain [map/satellite] images which may be copyrighted. All PLOS content is published under the Creative Commons Attribution License (CC BY 4.0), which means that the manuscript, images, and Supporting Information files will be freely available online, and any third party is permitted to access, download, copy, distribute, and use these materials in any way, even commercially, with proper attribution. For these reasons, we cannot publish previously copyrighted maps or satellite images created using proprietary data, such as Google software (Google Maps, Street View, and Earth). For more information, see our copyright guidelines: http://journals.plos.org/plosone/s/licenses-and-copyright.

 a. You may seek permission from the original copyright holder of Figure 3 to publish the content specifically under the CC BY 4.0 license. 

Response: Figure 3 is not copied from any source, it is produced by the authors in ArcGIS Pro software using the air pollution data downloaded from DEFRA office and the local authorities and LSOAs shape file for UK downloaded from the Office for National Statistics. Both data sets are available as open governmental data.

DEFRA air pollution data are open and publicly available from https://uk-air.defra.gov.uk/data/ under Open Government License (OGL), which allows anyone to copy, publish, distribute and transmit the information; adapt the information; exploit the information commercially and non-commercially for example, by combining it with other information, or by including it in own product or application. These data can therefore be used freely, as long as there is acknowledgement to DEFRA through citation.

The local authorities and LSOAs shape files for UK downloaded from the Office for National Statistics has an Open Government Licence and UK Government Licensing Terms and conditions of supply. The following statement is stated on their website (URL: https://www.ons.gov.uk/methodology/geography/licences ): “Under the terms of the Open Government Licence and UK Government Licensing Framework (launched 30 September 2010), if you wish to use or re-use ONS material, whether commercially or privately, you may do so freely without a specific application for a licence, subject to the conditions of the Open Government Licence and the Framework. If you are reproducing ONS content you must include a source accreditation to ONS. Digital boundary products and reference maps are supplied under the Open Government Licence. You must use the following copyright statements when you reproduce or use this material: Source: Office for National Statistics licensed under the Open Government Licence v.3.0”. 

We have added a footnote under Figure 3 to say that Figure 3 is produced by the authors using air pollution data from DEFRA and local authorities and LSOAs of UK shapefiles from the Office for National Statistics, both of which were downloaded under the Open Government Licence v.3.0. 

:

Additional Editor Comments:

We have received reviews from two reviewers for your manuscript “Air pollution and individuals’ mental well-being in the adult population in United Kingdom: A spatial-temporal longitudinal study and the moderating effect of ethnicity”. At this time, the manuscript will require substantial revision before it can be considered for publication. The reviewers recommended that you make substantial amendments to your manuscript. Please respond within the next 14 days to all comments raised by the reviewers. You can also submit a revised version of your manuscript at that time. We encourage you to submit your documents with tracked changes to highlight the revisions.

Reviewers' comments:

Reviewer's Responses to Questions

Comments to the Author

1. Is the manuscript technically sound, and do the data support the conclusions?

Reviewer #1: Yes

Reviewer #2: Yes

2. Has the statistical analysis been performed appropriately and rigorously?

Reviewer #1: Yes

Reviewer #2: No

3. Have the authors made all data underlying the findings in their manuscript fully available?

Reviewer #1: No

Reviewer #2: Yes

4. Is the manuscript presented in an intelligible fashion and written in standard English?

Reviewer #1: No

Reviewer #2: Yes

5. Review Comments to the Author

Reviewer #1: This study is interesting because the authors utilized the datasets from a large cohort to investigate the associations between mental health outcomes and exposures to NO2, SO2, PM2.5, and PM10. The study is properly designed, and data analyses are adequate with new findings. The data interpretations and discussions are adequate.

Specific comments:

1. Line 99: Please break this sentence into two separately focusing impacts of PM1 and PM2.5. A part of PM2.5 cannot cross blood brain barriers.

Response: We have added 2 sentences on line 93 to explain the specific impacts of PM2.5 and PM1 as follows: “Specifically, exposure to ambient PM2.5 results in depressive responses and increased hippocampal pro-inflammatory cytokines. (11) Exposure to PM1 leads to increased inflammation and reactive oxygen species (ROS) generation and impacts learning and memory (12)”.

2. Fig 4 title: correct subscripts.

Response: The subscripts have been corrected in Fig 4 and its title

3. Line 209: Please provide a reference justifying that 1x1 km spatial resolution is sufficient for selected air pollutants for this type of association studies. Please add more information on this.

Response: We have added two references to support that 1*1 km spatial resolution is sufficient for air pollution modelling and we have provided more information on line 247 as follows: “These air pollution maps at 1x1 km resolution are modelled each year by DEFRA under the “Defra's Modelling of Ambient Air Quality (MAAQ) contract” and are used to provide policy support in the UK and to fulfil the UK's reporting obligations to Europe (51). The 1x1 km air pollution raster data are the finest spatial resolution data that can be downloaded from DEFRA and are sufficient to obtain good modelling estimates (52, 53)”.

4. Lines 224 – 234: Besides citing references please also add some information stating the connections between these factors and targeted pollutants and health outcomes.

Response: We have added an explanation about the connections between the covariates and air pollution and mental well-being outcomes on line 294 as follows: “Specifically, poor mental well-being, stress, and depression have been associated with younger or older ages, women sex, cigarette smoking, alcohol drinking, physical inactivity, lower education, divorced/widowed marital status, lower household income, and belonging to an ethnic minority group (6, 31-33), which in turn confounds the association between air pollution and mental well-being outcomes”. 

5. Table 5: correct subscripts for air pollutants.

Response: The subscripts for air pollutants have been corrected in Table 5 which is now referred to as Table 6.

Reviewer #2: The authors tried to examine the effects of ambient air pollution on mental health by measuring mental well-being by a suitable tool. I also agree with authors that evidence base of the association between air pollution and mental health across the globe is week. In that context this study is important and has good scientific merits. The largest advantage of the study is the large sample size with repeated measure over 11 years. Though exposure misclassification is an issue as authors highlighted as population exposure assigned to individual, but that was only available option to them. However, there are lot of methodological issues, unless that are addressed, I cannot recommend for publication. These are the following issues:

Response: We have asked the data owner again to give us access to the lowest possible geography they have which is the Lower Super Output Areas (LSOAs) which is used to divide areas in the UK based on population size with the minimum allowed population size being 1000 people. Therefore, the paper now presents analysis at the LSOAs as well as the local authorities level. Results remained the same as we show in the paper, however, this approach reduces the exposure bias. 

We have also addressed the analysis issues as shown below:

1. I couldn’t understand why they dichotomized each item of GHQ12 tools. Instead, they could have added all considering each item measured in an ordinal scale of 4 points to have better precision in measurement. By doing so they reduced the true variability mechanical which led to decrease in precision of the measurement to some extent.

Response: There are 2 methods used in the literature to calculate the GHQ12 scale scores. The most used method is the (0-0-1-1) which dichotomises first the responses of individuals to each of the 12 questions and then sum them up resulting in an overall score that ranges between 0 and 12. The second method sums directly the scores (0-1-2-3) resulting in an overall score that ranges between 0 and 36. In the UKHLS data they have both methods. However, given that most of the literature uses the (0-0-1-1) method, we have used this. 

Nevertheless, to address the decrease in precision issue, we have now used both types of GHQ12 scale in our analysis and added an in-depth explanation under the “Individuals’ reported mental well-being” section starting on line 222-242 regarding the GHQ12 scores generation methods and how we used it in our study. All changes to this section are marked in Tracked changes. 

2. The most serious flaw of the analysis is not to report the distribution of the response of interest. As far as existing literature is concern that is supposed to be a negatively skewed distribution. Though I don’t know what their distribution was looked like, but my guess it is too both side truncated(bounded) negatively skewed distribution. Whatever means and sds reported by them in the discussion indicated presence of severe skewness. Question is how valid a Gaussian Mixed Effects Model is under this background of response distribution. No such attempt was made by the authors to check such validity of the multilevel model they used. Without that one cannot rely on their estimates or findings.

Response: The responses to the GHQ12 scale whether in terms of (0-12) or (0-36) scores were right skewed. After thorough review of the literature that used GHQ12 scale in their analysis, we have decided to do the analysis based on dichotomisation using cut-off values suggested by the literature to reflect the most accurate estimates of the GHQ12 scale in measuring the mental well-being of individuals. Therefore, we have constructed three GHQ12 binary variables and performed analysis on them. All the details of the construction of these three GHQ12 binary outcome variables are described in detail under the “Individuals’ reported mental well-being” section and marked in Track changes in the manuscript. 

We should note that the analysis for the three GHQ12 outcomes were concordant and that these results were also concordant with the previous analysis carried in the older version of this manuscript. Therefore, conclusions remained the same.

3. I recommend a Beta-Binomial mixed effects model to be used for this analysis which is commonly used by people with tool-based measurement of psychological features in the literature. Authors can convert the score into a proportion scale (0,1) by minimax normalization method. If any score attends exactly 0, or 1 do the necessary step to shrink them within (0,1). Report the results in terms OR per 10ug/m3 increase in pollutant. OR should be interpreted as the ratio of the odds of poor mental health wellbeing for every 10ug/m3 increase in ambient concentration of the pollutant. Please go through the relevant literature will guide you. As data wouldn’t be binary, the interpretation in terms of OR is a bit tricky.

Response: As reported in the previous response, we changed our analysis method to construct three binary GHQ12 outcomes rather than using the GHQ12 scale as a continuous variable following the methods of multiple published articles that used the GHQ12 questionnaire as detailed in the “Individuals’ reported mental well-being” section of the manuscript as follows: “For the present study, we used both methods for the GHQ12 scale, the (0-0-1-1) and the (0-1-2-3) method. Given that the scores of the GHQ12 scale are right-skewed and based on relevant literature, we dichotomised the overall GHQ12 scale using two cut off points for the GHQ12 (0-12): our sample mean GHQ12 (0-12) score = 1.8 ∼ 2 (GHQ0-12 ≥ 2) and the GHQ12 (0-12) score of greater than or equal to 4 (GHQ0-12 ≥ 4) as an indication of poor mental well-being (Godinho et al., 2011; Holi et al., 2003; Hori et al., 2016; Maheswaran et al., 2015; Nerdrum et al., 2006; Puustinen et al., 2011). The GHQ12 (0-36) score was dichotomised based on one cut off point of greater than or equal to 12 (GHQ0-36 ≥ 12) as an indication of poor mental well-being (Berglund et al., 2015; Wrede et al., 2021).”

Therefore, the three GHQ12 outcomes are now binary variables and not continuous variables as in the previous version of the manuscript. Thus, the beta-binomial method is not applicable anymore. Instead, we used for the analysis of these three GHQ12 binary outcomes multilevel mixed effects logit models, which are normally used by the literature to analyse binary variables. Please see below a list of some papers that used GHQ12 as a dichotomised binary variable and used logit models for analysis. 

Results in our paper are now reported in terms of ORs and 95% CIs per 10 µg/m3 increase in air pollutants to fit the used logit models.

We must note that we didn’t keep the analysis of the GHQ12 scale as a continuous variable and thus we didn’t use the reviewer suggested beta-binomial method because all relevant literature that uses the GHQ12 scale use it as a binary variable based on cut off thresholds validated by a large body of research studies to show poor mental well-being as we explained in the new version of the manuscript. We have also included below a list of research studies that use GHQ12 dichotomisation methods and logit models for analysis.

Reference list that used dichotomization of the GHQ12 scales and assessed associations using logit models: 

Berglund, E., Lytsy, P., & Westerling, R. (2015). Health and wellbeing in informal caregivers and non-caregivers: a comparative cross-sectional study of the Swedish general population. Health and Quality of Life Outcomes, 13(1), 109. https://doi.org/10.1186/s12955-015-0309-2

Godinho, E. L., Farias, L. C., Aguiar, J. C., Martelli-Júnior, H., Bonan, P. R., Ferreira, R. C., De Paula, A. M., Martins, A. M., & Guimarães, A. L. (2011). No association between periodontal disease and GHQ-12 in a Brazilian Police population. Med Oral Patol Oral Cir Bucal, 16(6), e857-863. 

Holi, M. M., Marttunen, M., & Aalberg, V. (2003). Comparison of the GHQ-36, the GHQ-12 and the SCL-90 as psychiatric screening instruments in the Finnish population. Nord J Psychiatry, 57(3), 233-238. https://doi.org/10.1080/08039480310001418

Hori, D., Tsujiguchi, H., Kambayashi, Y., Hamagishi, T., Kitaoka, M., Mitoma, J., Asakura, H., Suzuki, F., Anyenda, E. O., Nguyen, T. T. T., Hibino, Y., Shibata, A., Hayashi, K., Sagara, T., Sasahara, S., Matsuzaki, I., Hatta, K., Konoshita, T., & Nakamura, H. (2016). The associations between lifestyles and mental health using the General Health Questionnaire 12-items are different dependently on age and sex: a population-based cross-sectional study in Kanazawa, Japan. Environmental Health and Preventive Medicine, 21(6), 410-421. https://doi.org/10.1007/s12199-016-0541-3

Maheswaran, H., Kupek, E., & Petrou, S. (2015). Self-reported health and socio-economic inequalities in England, 1996–2009: Repeated national cross-sectional study. Social Science & Medicine, 136-137, 135-146. https://doi.org/https://doi.org/10.1016/j.socscimed.2015.05.026

Nerdrum, P., Rustøen, T., & Rønnestad, M. H. (2006). Student Psychological Distress: A psychometric study of 1750 Norwegian 1st‐year undergraduate students. Scandinavian Journal of Educational Research, 50(1), 95-109. https://doi.org/10.1080/00313830500372075

Puustinen, P. J., Koponen, H., Kautiainen, H., Mäntyselkä, P., & Vanhala, M. (2011). Psychological distress measured by the GHQ-12 and mortality: a prospective population-based study. Scand J Public Health, 39(6), 577-581. https://doi.org/10.1177/1403494811414244

Wrede, O., Löve, J., Jonasson, J. M., Panneh, M., & Priebe, G. (2021). Promoting mental health in migrants: a GHQ12-evaluation of a community health program in Sweden. BMC Public Health, 21(1), 262. https://doi.org/10.1186/s12889-021-10284-z

4. After understanding the data structure, I feel multilevel model may not be needed. There is no reason to believe that the temporal effects will vary over geography. Both temporal and cluster effects can be assumed independent considering two independent random intercepts. Secondly, the levels should be finalized by testing the variances step by step (H0: sigma_b=0 vs H1:sigma_b!=0). If nested model, then nested variances should be tested. Look at random effects model literature. ICC based assessment may not be correct always as that ignores the nested structure.

Response: The models now have the following nested structure in random intercept: repeated individual responses nested in Local authorities or LSOAs (i.e. two random intercepts in the models, one for the individual ID and one for the geography level of local authority or LSOAs). 

A multilevel model is needed for our analysis due to the nested structure of the data. As we explain in the manuscript, our data is made up of repeated individual responses over 10 or less data collection waves. Hence there is a need to account the models for the individual ID in random intercept. 

Second, individuals are nested within geographical areas (local authorities or LSOAs) and also air pollution is linked at a geographical scale (once at the local authority level and once at the LSOAs level). Therefore, there is a need for another random intercept for the geographical scale (local authority or LSOA) to account for the nested structure of the data and obtain accurate estimates of the effect of air pollution on mental well-being. 

In constructing our multilevel models, we didn’t just rely on the ICC but also on the nested structure of the data and the contextual linkage of air pollution. The ICC was used just to show that accounting for individuals ID in random intercepts is necessary given the high observed ICC while that of the household ID might not be needed given the low observed ICC. 

We have added into the manuscript on line 335-343 an explanation of why a multilevel model is needed supported by a reference from the literature as follows: “This type of analysis was chosen as it fits the longitudinal panel design of the study which involves repeated individual responses across time linked to air pollution data at the LSOA or local authority level whereby repeated individual responses are nested within LSOAs or local authorities. The random intercept for the individual ID is necessary in the multilevel models given the high homogeneity in the individual’s responses across time (ICC=0.42 and ICC=0.49; Table 5), while the local authority or LSOAs random intercept is needed to allow for less biased assessments of the contextual-linked air pollution effect on mental well-being (Jerrett et al., 2014); resulting in three-levels mixed-effect logit models”.

Jerrett, M., McConnell, R., Wolch, J., Chang, R., Lam, C., Dunton, G., Gilliland, F., Lurmann, F., Islam, T., & Berhane, K. (2014). Traffic-related air pollution and obesity formation in children: a longitudinal, multilevel analysis. Environmental Health, 13(1), 49. https://doi.org/10.1186/1476-069X-13-49

5. I am confused with within vs between effects of pollutants. After reading their process of computation I understand they all are same measured with different level of precision. 1) manually removed the temporal variation and estimated effect size 2) manually removed spatial variation and estimated effect size 3) statistically removed spatiotemporal variation and estimated effects size. Where are the differences? First fix the questions you are asking. The possible questions could be 1) Do the effects of pollution vary over geography? 2) Do the effects vary over duration of exposure? Stratification by interaction term should be the solution of it.

Response: The between effects are the average air pollution exposure across the 11 years of follow-up. Therefore, the measure can provide evidence of whether more polluted local-authorities or LSOAs affect the mental well-being of individuals. 

The within effect is the annual deviation of air pollution from the average 11 years air pollution per each local authority or LSOA; thus this measure shows whether fluctuations of air pollution across time within each local authority or LSOA affect individuals’ mental well-being. For example, if person A has poorer well-being than person B – this could be any for a reason/characteristic which make them different; Person A (or B) well-being might become poorer if air pollution conditions change from one year to another, keeping all other conditions/variables fixed: this is the within effects. 

We have amended the research questions on line 126-135 and on line 157-170 to reflect the analysis done as follows: “Accordingly, this study investigates longitudinally the overall and the spatial-temporal (between-within) effects of long-term (11 years) exposure to NO2, SO2, PM10, and PM2.5 air pollution in the UK on individuals’ reported mental well-being measured using the 12 items General Health Questionnaire (GHQ12) scale. Unlike other studies that assess the effect of air pollution on well-being using one geographical scale, our study aims to assess the effect of air pollution exposure on mental well-being at two geographical scales, coarse local authorities (council areas in Scotland) and detailed Lower Super Output Areas (LSOAs; data zones in Scotland). This will allow us to compare the results between the two geographical scales and explore more deeply the local-contextual patterns of the effect of air pollution on mental well-being. Additionally, our study aims to consider both space and time by determining whether living in more polluted geographical areas (local authorities and LSOAs) is the driving cause for poor mental well-being (between) or whether it is the variation in air pollution across time within each geographical area (within) that is causing poor mental well-being; thus providing detailed spatial-temporal evidence for policymaking decisions”. 

We have also added two equations under the data analysis section on line 361-373 to explain the between-within effects in the multilevel mixed effects logit models. 

6. Supplementary analysis should be done by either baseline logistic model or ordered logistic, not by binary logistic model. By multiple binary logistic models’ precision of the estimates would be compromised.

Response: This supplementary material has now been removed, and multilevel mixed effects logit models have been moved into the main paper as per the response to points 2 and 3 above. The three GHQ12 outcomes are binary outcomes coded as 0=good mental well-being and 1=poor mental well-being, therefore we have used for analysis multilevel mixed effects logit models as per the relevant literature that uses the same GHQ12 scale measure. All changes are marked in Track changes in the manuscript. 

7. In the abstract authors reported regression coefficients but named as OR, I guess. As higher score means worst in mental wellbeing, one should expect a positive association with pollution, OR should be >1.

Response: We apologise for this typo mistake. However, given that the analysis is changed and is now performed on three GHQ12 binary outcomes using multilevel mixed effect logit models, we are reporting in this new version of the abstract ORs not coefficients. 

8. Line 385-389: Authors stating in text about high score but reporting regression coefficients doesn’t go with sentence. Is it the mean difference from reference ethnic group? If so, state accordingly.

Response: The sentence was corrected on line 577 after carrying out the new analysis of the three GHQ12 binary outcomes as follows: “It is worth to note that examining the association between ethnicity and mental well-being revealed higher odds of poor mental well-being among people from Pakistani/Bangladeshi (GHQ0-36 ≥ 12: OR=1.17, 95%CI=1.06-1.29; GHQ0-12 ≥ 2: OR=1.09, 95%CI=1.00-1.19; GHQ0-12 ≥ 4: OR=1.13, 95%CI=1.02-1.25) and mixed ethnicities (GHQ0-36 ≥ 12: OR=1.15, 95%CI=1.01-1.32; GHQ0-12 ≥ 2: OR=1.21, 95%CI=1.08-1.37; GHQ0-12 ≥ 4: OR=1.26, 95%CI=1.10-1.43) origin in comparison to the British-white, even after adjusting for socio-demographics and lifestyle covariates (S1 Table 1)”. 

9. Looking at the correlation matrix among pollutant, it is worthy to try multipollutant model with SO2, NO2 and PM2.5. Look at variance inflation factor, if less than 8/10, don’t bother, it will work. Best way to assess is add one at a time if AIC decreases you can retain. In case of multicollinearity AIC should increase after adding them in a single model.

Response: Given that the correlation between SO2 and the other three pollutants is low to moderate, we have added to our analysis bi-pollutant models adjusting the NO2, PM10, and PM2.5 models to SO2 pollutant. The results of this analysis are shown in Table 7 and Table 9 in the manuscript.

---

## [Editor Report · Decision Letter 1]

10 Feb 2022

Air pollution and individuals’ mental well-being in the adult population in United Kingdom: A spatial-temporal longitudinal study and the moderating effect of ethnicity

PONE-D-21-31292R1

Dear Dr. Abed Al Ahad,

We’re pleased to inform you that your manuscript has been judged scientifically suitable for publication and will be formally accepted for publication once it meets all outstanding technical requirements.

Kind regards,

Bijaya Kumar Padhi, PhD, MPH

Academic Editor

PLOS ONE
---

## [Editor Report · Acceptance letter]

15 Feb 2022

PONE-D-21-31292R1 

Air pollution and individuals’ mental well-being in the adult population in United Kingdom: A spatial-temporal longitudinal study and the moderating effect of ethnicity 

Dear Dr. Abed Al Ahad:

I'm pleased to inform you that your manuscript has been deemed suitable for publication in PLOS ONE. Congratulations! Your manuscript is now with our production department. 

Kind regards, 

on behalf of

Dr. Bijaya Kumar Padhi 

Academic Editor

PLOS ONE